# Neural and computational underpinnings of biased confidence in human reinforcement learning

Chih-Chung Ting [1,2] ✉, Nahuel Salem-Garcia [3], Stefano Palminteri [4,5], Jan B. Engelmann [2,6,8] ✉ & Maël Lebreton [3,7,8] ✉

While navigating a fundamentally uncertain world, humans and animals constantly evaluate the probability of their decisions, actions or statements being correct. When explicitly elicited, these confidence estimates typically correlates positively with neural activity in a ventromedial-prefrontal (VMPFC) network and negatively in a dorsolateral and dorsomedial prefrontal network. Here, combining fMRI with a reinforcement-learning paradigm, we leverage the fact that humans are more confident in their choices when seeking gains than avoiding losses to reveal a functional dissociation: whereas the dorsal prefrontal network correlates negatively with a condition-specific confidence signal, the VMPFC network positively encodes task-wide confidence signal incorporating the valence-induced bias. Challenging dominant neuro-computational models, we found that decision-related VMPFC activity better correlates with confidence than with option-values inferred from reinforcement-learning models. Altogether, these results identify the VMPFC as a key node in the neuro-computational architecture that builds global feeling-of-confidence signals from latent decision variables and contextual biases during reinforcement-learning.

Humans and animals seem to be constantly engaged in computing the subjective probability of having made the right choice, having successfully memorized or recognized a cue, having correctly executed the desired action or having endorsed the most truthful statement, which can typically be explicitly elicited as confidence judgments[1–5]. These metacognitive confidence judgments are increasingly considered as having a critical functional role in (sequential) decision-making, controlling the integration of new evidence[6], adjusting speed-accuracy trade-offs[7], and triggering changes of mind[8,9]. Likewise, a

recent but increasing number of studies suggests that confidence could be a key variable to understand human (reinforcement-) learning behavior both at the normative and descriptive levels[10–15].

At the neurobiological levels, the computation of confidence and the production of confidence judgments has been consistently associated with neural activity in two main prefrontal networks across a large variety of cognitive tasks: a negative prefrontal network, encompassing dorsal anterior cingulate cortex (dACC), bilateral insula, dorso-medial and dorsolateral prefrontal cortices, and a positive

[1]General Psychology, Universität Hamburg, Von-Melle-Park 11, 20146 Hamburg, Germany. [2]CREED, Amsterdam School of Economics (ASE), Universiteit van Amsterdam, Roetersstraat 11, 1018 WB Amsterdam, the Netherlands. [3]Swiss Center for Affective Science, Faculty of Psychology and Educational Sciences, University of Geneva, Chem. des Mines 9, 1202 Genève, Switzerland. [4]Département d'Études Cognitives, École Normale Supérieure, PSL Research University, 29 rue d'Ulm, 75230, Paris cedex 05, France. [5]Laboratoire de Neurosciences Cognitives et Computationnelles, Institut National de la Santé et de la Recherche Médicale, 29 rue d'Ulm 75230, Paris cedex 05, France. [6]The Tinbergen Institute, Gustav Mahlerplein 117, 1082 MS Amsterdam, the Netherlands. [7]Economics of Human Behavior group, Paris-Jourdan Sciences Économiques UMR8545, Paris School of Economics, 48 Boulevard Jourdan, 75014 Paris, France. [8]These authors contributed equally: Jan B. Engelmann, Maël Lebreton. ✉e-mail: chihchung.ting@uni-hamburg.de; j.b.engelmann@uva.nl; mael.lebreton@psemail.eu

ventral network, mostly centered around the ventromedial prefrontal cortex[16–20]. For instance, dACC was originally identified as a key center for performance monitoring and error detection[21,22] as well as for the computation of uncertainty-related variables[23], before being more generally integrated as a part of a large network negatively correlating with confidence judgments[17,18,24–26]. More recently, BOLD activity in the ventromedial prefrontal cortex (VMPFC) and pregenual anterior cingulate cortex (pgACC) has been positively associated with confidence and self-performance evaluation, first in the context of value-based decision-making[27], and then more broadly in other contexts and tasks[3,17,18,24,28,29].

While both positive and negative prefrontal networks are omnipresent in the most recent meta-analyses and theories of confidence and metacognition judgments[16,19] there is, to date, very little empirical evidence to formally dissociate the relative roles of those two networks in the computation of confidence—but see e.g.,[16,24]. One promising hypothesis is that some of those network elements could be involved in different stages of confidence processing, including computing and integrating different confidence-building variables such as levels of uncertainty. Uncertainty and confidence can indeed be distinguished at the theoretical and computational levels: while confidence can be defined as the probability that a decision (or a proposition) is correct given the evidence, (un)certainty refers to the encoding of all other probability distributions over sensory and cognitive variables on which choices and confidence are ultimately built[1,4,16]. Thereby, these two quantities might be easily confoundable—potentially explaining why they have been associated with similar brain regions and neural patterns of activity in previous studies—but remain theoretically dissociable. Given the previous association of the negative network with uncertainty and error detection[5], and of the positive network with affect and subjective valuation[30], one credible neurocomputational architecture would ascribe to the negative network a role in representing objective uncertainty—which often (negatively) correlates with confidence—, and to the VMPFC a role in aggregating a composite variable corresponding to the subjective, phenomenological feeling of confidence, from decision-related uncertainty variables and all other incidental signals influencing confidence.

Here, to test this putative architecture, we leverage a reinforcement learning paradigm that naturally orthogonalizes specific dimensions of difficulty and affective information (Fig. 1a, b), by factorially manipulating two features of choice outcomes: their valence (monetary gains or losses) and the quantity of information (partial versus complete feedback). Our idea is to take advantage of the valence-induced bias in confidence judgments described in the context of this task—i.e., the fact that participants are genuinely more confident in their choices when seeking gains than avoiding losses, despite identical objective difficulty and learning performance[31–33] (Fig. 1c). Considering the task features and the typical participant behavior, a brain region encoding *objective uncertainty* should therefore correlate with confidence in all conditions, and exhibit signal differences between complete and partial-information contexts, as the objective uncertainty is higher in partial than complete-information contexts. On the other side of the spectrum, a brain region encoding *task-wide confidence* (corresponding to the reported, absolute feeling of confidence) should correlate with confidence in all conditions, and exhibit signal differences between gain and loss contexts, as participants report higher confidence in a gain context (despite similar choice difficulty and performance observed in a loss context). Finally, we also define a third variable, *condition-specific confidence*, which simply indexes the relative increase of confidence in each learning context due to the incremental improvement of choice accuracy caused by feedback-based learning. A brain region encoding condition-specific confidence should therefore correlate with confidence in all contexts, but not exhibit any signal difference due to our manipulation of valence and information (Fig. 1d).

Following this reasoning, we recorded BOLD activity in participants while they performed the reinforcement-learning task featuring manipulations of outcome valence and information quality, paired with confidence elicitations. Behavioral analyses first confirmed the presence of the valence-induced confidence bias. fMRI analyses showed that confidence was positively and negatively related to the activity in the prefrontal networks regardless of affective information and task difficulty manipulations. Using theory-driven qualitative patterns of activation as well as a quantitative model comparison exercise, our neuro-imaging analyses then revealed a functional dissociation. On the one hand, neural activity in the negative prefrontal network (i.e., DMPFC and DLPFC) correlated with a condition-specific confidence signal that gradually builds up, independently in each learning context. On the other hand, neural activity in the positive prefrontal network (i.e., VMPFC) additionally integrates contextual effects such as the valence-induced confidence bias, thereby representing absolute, task-wide confidence that mimics the feeling-of-confidence reported by participants. We further verified the role of the positive network in reinforcement learning via model-based fMRI analysis. In short, while VMPFC was also engaged in the computational process, the activity in the VMPFC can be better explained by confidence than other ongoing computational variables, including chosen option values and value differences.

## Results

Forty participants took part in our experiment and completed the instrumental learning task in the MRI scanner. During the learning task (Fig. 1a), participants repeatedly faced pairs of abstract symbols (cues), that were probabilistically associated with monetary outcomes (gains or losses). In each pair, also referred to as context, one cue was associated with a better-expected outcome (i.e., higher probability of gain or lower probability of loss), and the goal of participants was to learn, by trial and error, to identify and preferentially choose this cue. Two main contextual factors were orthogonally manipulated: outcome valence and outcome information[31,33,34]. The valence factor defines Gain and Loss contexts, which respectively only include cues probabilistically associated with gains or losses (Fig. 1b). The information factor defines Partial and Complete information contexts, where feedback is respectively provided only for the chosen cue, or for both the chosen and unchosen cues (Fig. 1c). In addition, at each trial, participants reported their confidence in their choice on a probabilistic scale as the subjective probability of having made a correct choice from 50% indicating chance level to 100% (indicating certainty). Those confidence judgments were incentivized using a matching probability mechanism—see "Methods" and refs. 35, 36 for details. Note that we decoupled the decision and response-related processes by delaying the mapping between the cue and the motor response, so as to minimize the inherent correlation between decision response times and confidence judgments—see Fig. 1a, "Methods" and ref. 33 for details.

### Reinforcement-learning behavior features the valence-induced confidence bias

Overall, participants' choice accuracy (i.e., the average probability of choosing the better symbol) is above guessing level ($t_{39} = 17.78$; $P < 0.001$; Supplementary Table S1), indicating that they were able to identify and select the better symbols from the probabilistic outcomes, by trial and error. We then evaluated the effects of our main experimental factors on the two behavioral variables of interest: choice accuracy and confidence judgments (Fig. 2). Replicating previous reports[31–34,37], we confirmed that choice accuracy is modulated by information but not valence (two-way repeated-measures ANOVA; valence: $F_{1,39} = 0.00$, $P = 0.9666$; information: $F_{1,39} = 22.05$, $P < 0.001$; interaction: $F_{1,39} = 0.01$, $P = 0.9056$).

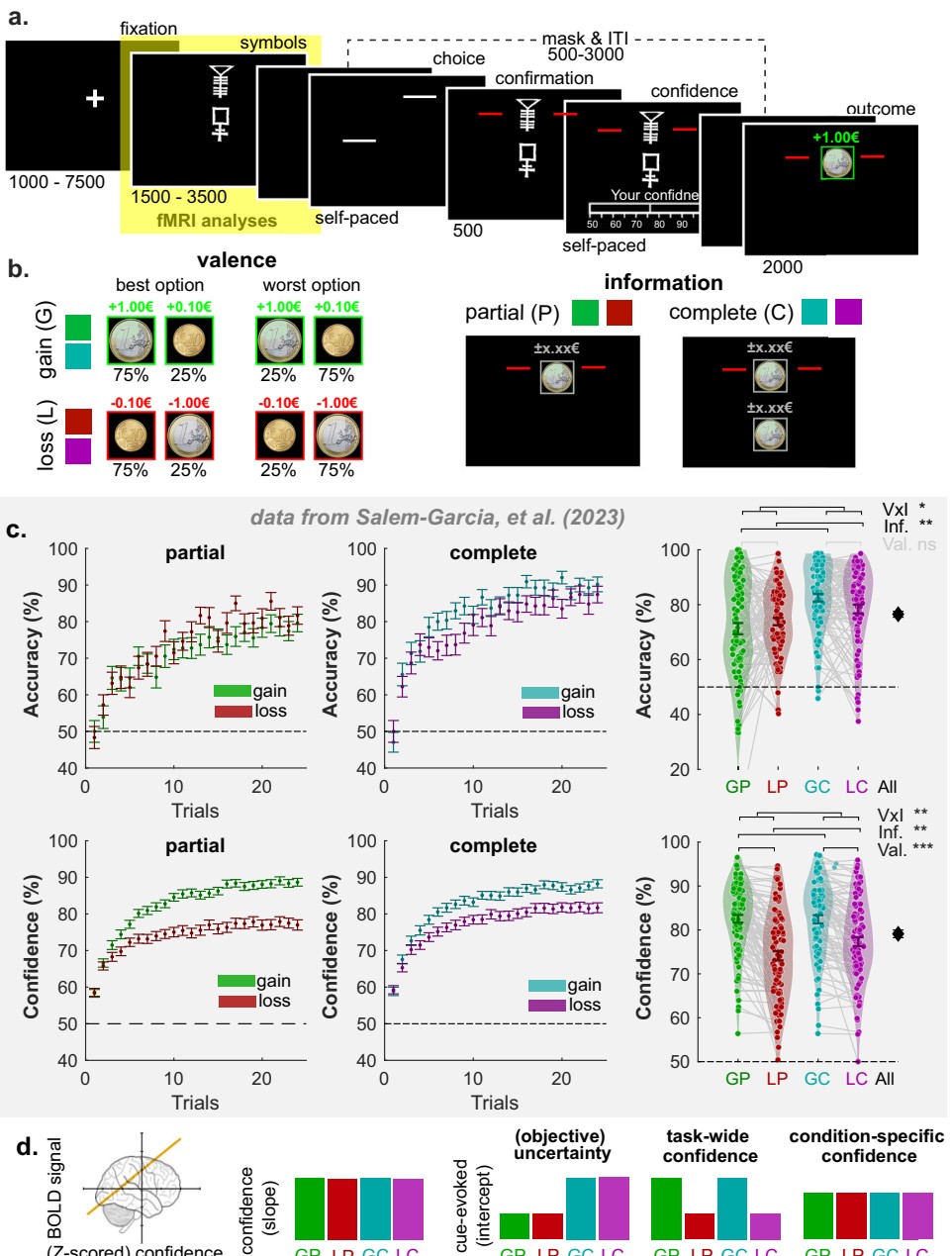

**Fig. 1 | Experimental design and hypotheses. a** Successive screens displayed during the learning task. Durations are given in ms. The yellow box highlights the event of interest for the fMRI analyses. **b** Illustration of two-by-two factorial design with outcome valence (gain and loss) and information (partial and complete) manipulations. Each condition is consistently associated with a pair of symbols in each run. Each symbol is consistently associated with a probability (75% or 25%) of getting larger gains (€+1.0) and smaller gains (€+0.1) in the gain conditions and is consistently associated with a probability of getting smaller losses (€−0.1) and larger losses (€−1.0) in the loss conditions. In the outcome phase, the outcome from the chosen symbol is always displayed and highlighted with two red bars regardless of the information condition. The outcome from the unchosen option is absent in the partial information condition but is available in the complete information condition. **c** Evolution of average accuracy (upper panels) and confidence

(bottom panels) across trials from five instrumental learning tasks ($n = 90$ independent human participants from five experiments), which were reanalyzed and reported in ref. 32. Different colors represent different contexts following the conventions from (**a**). Dots and error bars represent the trial-resolved mean ± SEM of the participant data. **d** Qualitative predictions about the relationship between brain activation patterns (BOLD signal) and confidence (e.g., the yellow line), for three possible confidence-related signals: uncertainty, condition-specific confidence, and task-wide confidence. The relationships can be summarized with a slope and an intercept (cue-evoked), across conditions. GP gain/partial, LP loss/partial, GC gain/complete, LC loss/complete, Val. Valence manipulation, Inf. Information manipulation, V × I Valence and information interaction. -: $0.05 < P < 0.1$; *: $0.01 < P < 0.05$; **: $0.001 < P < 0.01$; ***: $P < 0.001$.

Again replicating previous reports[31–33], our analysis confirmed that confidence, on the other hand, is additionally affected by valence (valence: $F_{1,39} = 36.56$, $P < 0.001$; information: $F_{1,39} = 6.76$, $P = 0.0131$; interaction: $F_{1,39} = 9.62$, $P = 0.0036$). In addition to confidence being generally higher in gain than loss contexts, this valence effect was larger in the partial than in the complete information condition (post-hoc t-tests; partial: $t_{39} = 6.93$, $P = 2.68 \times 10^{-8}$: complete: $t_{39} = 4.55$, $P = 5.08 \times 10^{-5}$; difference: $t_{39} = 3.10$, $P = 0.0451$; Fig. 2b). Overall, these results confirmed the presence of a valence-induced bias in confidence judgments that is mitigated by complete information.

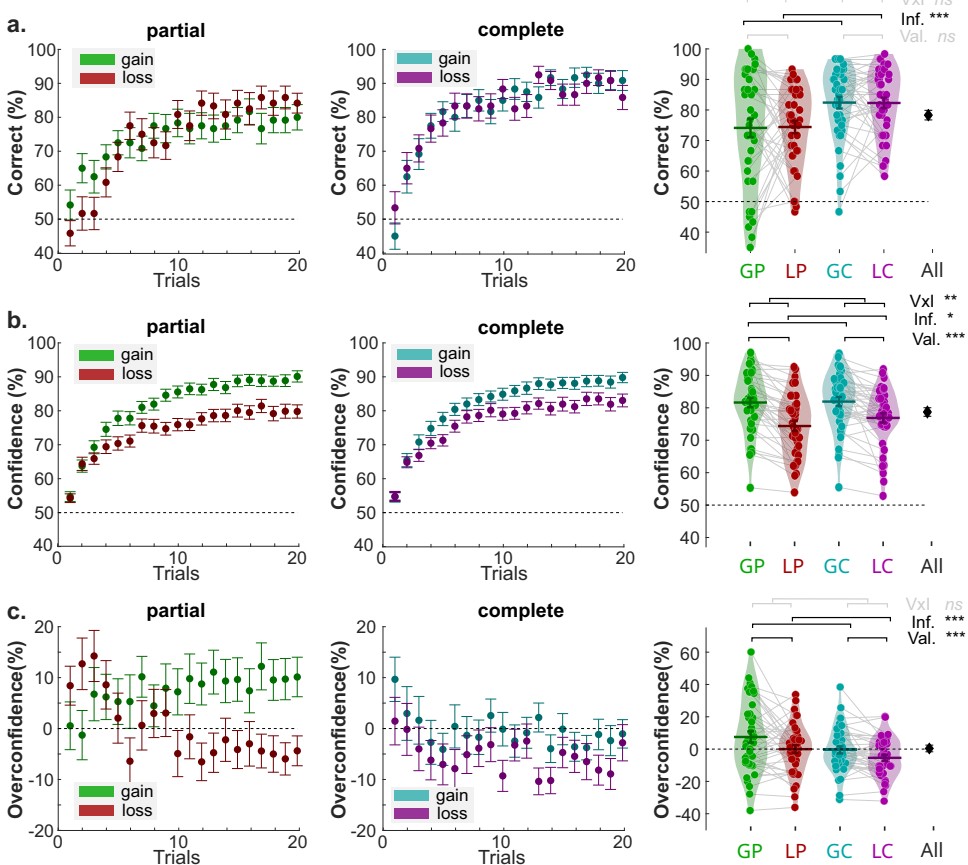

**Fig. 2 | The effect of outcome valence and information on learning and confidence.** Left and middle panels are trial-by-trial (**a**) percentage of correct responses, **b** Confidence rating, and (**c**) overconfidence in the partial information (left panels) and complete information condition (middle panels). Dots and error bars represent the trial-resolved mean ± SEM of the participant data. Right panels picture condition-specific averages. **a** Percentage of correct responses, **b** confidence rating, and **c** overconfidence across conditions at the individual level (colored dots; $n = 40$ independent participants) and group-level (horizontal bars). Two-way repeated-measures ANOVAs indicated that choice accuracy is modulated by information but not valence (valence: $F_{1,39} = 0.00$, $P = 0.9666$; information:

$F_{1,39} = 22.05$, $P < 0.001$; interaction: $F_{1,39} = 0.01$, $P = 0.9056$), while confidence is additionally affected by valence (valence: $F_{1,39} = 36.56$, $P < 0.001$; information: $F_{1,39} = 6.76$, $P = 0.0131$; interaction: $F_{1,39} = 9.62$, $P = 0.0036$). As a result, calibration was modulated by valence and information (two-way repeated-measures ANOVA; valence: $F_{1,39} = 12.28$, $P = 0.0012$; information: $F_{1,39} = 14.42$, $P < 0.001$; interaction: $F_{1,39} = 0.58$, $P = 0.4506$). The black error bars indicate the overall performance over conditions. The colored horizontal bar and error bar represent the mean and SEM, respectively. Val Valence; Inf. Information. V × I interaction between Valence and Information. -: $0.05 < P < 0.1$; *: $0.01 < P < 0.05$; **: $0.001 < P < 0.01$; ***: $P < 0.001$.

We also contrasted confidence and choice accuracy to properly characterize overconfidence (or calibration). On average, calibration was non-significantly different from 0, indicating neither over- nor under-confidence ($t_{1,39} = 0.1883$, $P = 0.8516$). Yet, replicating previous finding[31,32] we found that participants were significantly overconfident in the Gain-Partial context ($t_{1,39} = 2.14$, $P = 0.0385$), and that calibration was significantly modulated by valence and information, with Losses and Complete information improving calibration (valence: $F_{1,39} = 12.28$, $P = 0.0012$; information: $F_{1,39} = 14.42$, $P < 0.001$; interaction: $F_{1,39} = 0.58$, $P = 0.4506$; Fig. 2c and Supplementary Table S2). These results held when we tested generalized linear mixed-effect models, in which we used trial-by-trial data and included predictors accounting for valence, information, the session number, and response times (Supplementary Table S3).

Finally, response times featured a small but significant residual effect of valence (valence: $F_{1,39} = 4.77$, $P = 0.0350$; information: $F_{1,39} = 0.31$, $P = 0.5782$; interaction: $F_{1,39} = 0.97$, $P = 0.3318$), as well as a negative correlation with confidence judgments (Supplementary Table S2). Despite the dissociation between decision and response processes, there was a significant correlation between response times and confidence judgments (Supplementary Table S4). Nevertheless, the valence-induced confidence bias and the valence-induced RT

effect were not correlated at the inter-individual level (robust regression slope: $\beta = -0.01 \pm 0.01$, $P = 0.339$). Moreover, an interindividual regression analysis suggested the valence-induced confidence bias could be observed in the absence of a valence-induced RT bias (robust regression intercept: $\beta = 5.02 \pm 0.84$; $P < 0.001$; Supplementary Table S5). These results are in line with our previous finding that the valence-induced bias on confidence and on RTs are partially dissociable[33].

## Confidence is encoded in a positive ventromedial-prefrontal and a negative parieto-frontal network

Our neuroimaging investigations focus on confidence signals that are elicited at the decision stage (i.e., during symbol presentation, in which a motor response is not required). First, we aimed to identify neural networks whose activity generally correlates with confidence judgments during option evaluation across learning contexts. To do so, we designed a first general linear model (GLM1), in which the cue presentation period was modeled separately in each of the four contexts, and each of these events was modulated by the time series of context-specific, trial-by-trial confidence judgments (see "Methods" and Table 1 for the complete GLM1 specification; note that, to ensure between-subject and between-regressor commensurability, all parametric

**Table 1 | GLMs' structure**

|  | Symbols | Choice | Confidence | Outcome |
|---|---|---|---|---|
| GLM1 | GP_onset ×GP_conf.<br>LP_onset ×LP_conf.<br>GC_onset ×GC_conf.<br>LC_onset ×LC_conf. | all_onsets ×choice (R/L) | all_onsets ×dist. | GP_onset ×GP_out.<br>LP_onset ×LP_out.<br>GC_onset ×GC_out<br>LC_onset ×LC_out. |
| GLM2$_{WID}$ | all_onsets ×all_conf. (nat.) | all_onsets ×choice (R/L) | all_onsets ×dist. | all_onsets ×all_out. |
| GLM2$_{SPE}$ | all_onsets ×all_conf. (Z/cond) | all_onsets ×choice (R/L) | all_onsets ×dist. | all_onsets ×all_out. |
| GLM3 | all_onsets<br>×Qc<br>×Qu<br>×V | all_onsets ×choice (R/L) | all_onsets ×dist. | all_onsets ×all_PE |
| GLM4 | all_onsets<br>×Qc<br>×\|Qc-Qu\|.<br>×conf. $_{t-1}$ | all_onsets ×choice (R/L) | all_onsets ×dist. | all_onsets ×all_PE |
| GLM5 | all_onsets<br>×Qc<br>×conf. | all_onsets ×choice (R/L) | all_onsets ×dist. | all_onsets ×all_PE |

The table represents the four events of interest in a trial as columns, and list for each GLM, the corresponding regressors and their respective parametric modulators (indicated by a × sign).
Parametric modulators, Qc, Qu, V, and PE, are estimated by the winning model.
For GLM3-5, which feature several parametric modulators on the same event, SPM's serial orthogonalization was turned off.
*GP* Gain-Partial, *LP* Loss-Partial, *GC* Gain-Complete, *LC* Loss-Complete, *conf* confidence; (R/L) choice coded as 1/−1 for right/left, *dist.* distance (difference between the starting point and final confidence rating), *out.* outcome (coded 1/0 if the chosen outcome is the best/worst potential outcome – i.e., 1 and −0.1 are encoded as 1 and 0.1 and −1 are encoded as −1), *Qc* chosen option values, *Qu* unchosen option value, *V* context value, *PE* prediction error.

modulators of all fMRI GLMs were z-scored at the session and individual level, see "Methods" and[38]). A random-effects analysis looking at BOLD signals that were correlated with the confidence parametric modulators across contexts identified two main brain networks (voxel-wise $P_{uncorrected} < 0.001$; cluster-wise $P_{FWE} < 0.05$; Fig. 3a and Supplementary Tables S6, S7). On the one hand, neural activity in the VMPFC, pgACC, precentral gyrus, and middle temporal gyrus correlated positively with confidence rating. On the other hand, activity in a large parieto-frontal network encompassing dorsolateral (bilateral IFG and INS) and dorsomedial prefrontal clusters (dACC and DMPFC) correlated negatively with confidence judgments. A small cluster in the left caudate also correlated negatively with confidence (see Supplementary Table S7). At the whole brain level, no brain region exhibited a valence or information effect on confidence encoding, nor an interaction between those factors (rmANOVA and direct contrasts).

To better characterize the signal encoded in the confidence-encoding prefrontal regions, we then regrouped the prefrontal clusters identified in our whole-brain analysis into three main functional regions(/networks)-of interest (ROIs), respectively representative of ventromedial (VMPFC), dorsolateral (DLPFC: union of bilateral INS and IFG) and dorsomedial (DMPFC, dACC) prefrontal cortices. Then, we extracted, in these ROIs, the parametric confidence regression coefficients for all four contexts. We first verified that our experimental manipulations of outcome valence and outcome information did not impact this parametric encoding of confidence (all $Ps > 0.05/3$; Supplementary Fig. S6b and Supplementary Tables S6, S7). No significant effect of those factors was found (Bonferroni-corrected for three comparisons). Overall, these analyses confirmed that VMPFC on the one hand, and DLPFC and DMPFC on the other, respectively constitute the positive and negative confidence-encoding networks, and that they encode confidence similarly across the different contexts.

**Task-wide vs. condition-specific confidence in the brain**
Next, we turned to our main question of interest, namely dissociating different types of confidence and uncertainty signals, which we ultimately hoped could help in identifying functionally dissociable brain networks. We defined three theoretical types of qualitative patterns on those cue-evoked activities, that specifically characterize three confidence-related neural signals: uncertainty, condition-specific

confidence and task-wide confidence (Fig. 1d). Essentially, statistical uncertainty corresponds to the objective difficulty of the choice, that is ultimately revealed in choice accuracy. Accordingly, statistical uncertainty should be higher in Partial than in Complete information contexts, but identical in Gain and Loss contexts, given the similar objective difficulty and observed performance between these conditions (Fig. 1d). Condition-specific confidence simply tracks the subjective, relative improvement within each context, and is reminiscent of the context value that tracks the choice-independent expected value in each context[34,39]. Thereby, condition-specific confidence should be purely context-dependent, hence not show any effect of our factors (Fig. 1d). Finally, task-wide confidence corresponds to the actual absolute, phenomenological feeling of confidence that is reported as the confidence judgments. Task-wide confidence should then be higher in Gain than Loss context, with potentially a mitigation by information (Fig. 1d). From those definitions, and given that our ROIs have already been shown to encode confidence across contexts, one can simply ascribe those theoretical variables to ROI activity, by testing the effect of valence and information on cue-evoked activity, as modeled in GLM1 (Fig. 1d). We found a significant valence effect ($F_{1,37} = 8.99$, $P = 0.0048$) and marginal valence-information interaction in VMPFC ($F_{1,37} = 3.99$, $P = 0.0532$) (Fig. 3b and Supplementary Table S6). Mimicking the pattern of confidence judgments, the difference between BOLD activity elicited in gain versus loss contexts was higher in the partial than in the complete information context (partial: $0.62 \pm 0.18$; complete: $0.14 \pm 0.16$; $t_{37} = 1.99$, $P = 0.0532$). In contrast, we did not find significant effects of the valence and information factors on BOLD activity in either of the negative networks ($Ps > 0.08$). The results of this ROI analysis tentatively ascribe task-wide confidence signals (including a valence effect) to the VMPFC and condition-specific confidence (without valence nor information effects) to both DLPFC and DMPFC. For completeness, we also tested for additional whole-brain activation for the positive and negative effects of valence and information on cue-evoked activity. The result revealed that only the Gain > Loss contrast elicited activations in a large brain network encompassing, among other regions, the VMPFC (voxel-wise $P_{uncorrected} < 0.001$; cluster-wise $P_{FWE} < 0.05$; Supplementary Fig. S6 and Supplementary Table S7). Finally, we performed a whole-brain conjunction between regions correlating positively with confidence

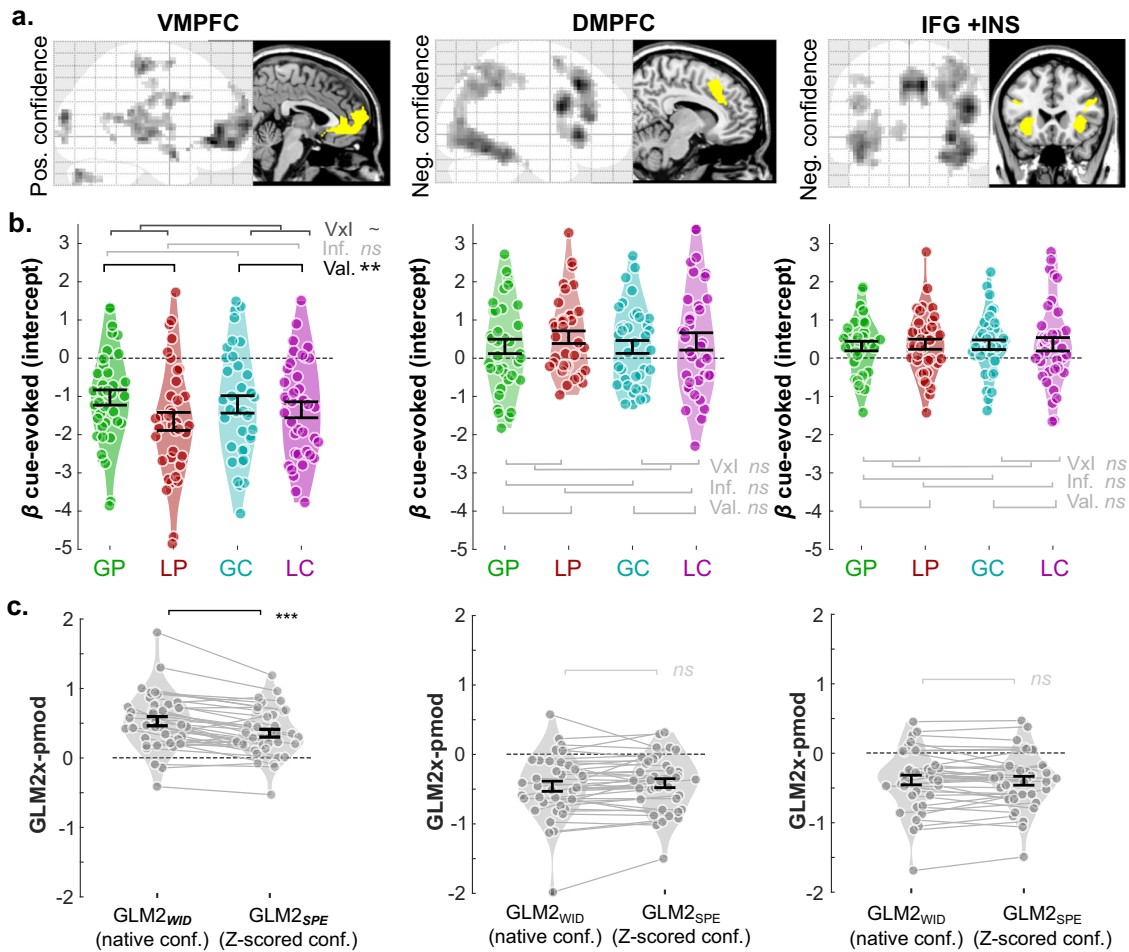

**Fig. 3 | Model-free fMRI results for the learning task. a** Results of whole-brain analysis. Brain areas positively (left panels) and negatively (middle and right panels) correlate with confidence rating during the symbol presentation phase. Significant voxels are displayed on the glass brains in a gray-to-black gradient manner (one-sided tests; $p_{uncorrect} < 0.001$, cluster size >47; cluster-wise $P_{FWE} < 0.05$). The yellow areas in the anatomical brain are ROIs (vmPFC, dmPFC, and IFG + INS), which are used in the following ROI analyses. **b** Violin plots represent the sample distribution of fMRI regression coefficients of cue-evoked signals for the different contexts (represented by different colors). Note that the notion of positive versus negative network characterizes the sign of the correlation of activations with confidence. In the present panel, cue-evoked activity exhibits an opposite pattern, with negative baseline activations in the positive network, and positive baseline activations in the negative network. Dots correspond to individual regression coefficients ($n = 38$ independent participants). Error bars represent sample mean ± SEM. GP gain/partial, LP loss/partial, GC gain/complete, LC loss/complete. Two-way repeated-measures ANOVAs indicated that only VMPFC cue-evoked activation is affected by our experimental manipulation, with a significant valence effect and a marginal valence-information interaction (valence: $F_{1,37} = 8.99$, $P = 0.0048$; interaction: $F_{1,37} = 3.99$, $P = 0.0532$). **c** Violin plots represent the sample distribution of fMRI regression coefficients for native versus Z-scored confidence regression coefficients, respectively extracted from GM2$_{WID}$ and GLM2$_{SPE}$. Paired $t$ tests indicated that regression coefficients for native confidence are significantly higher than for Z-scored confidence in the VMPFC (two-sided tests; $t_{37} = 5.41$, $P < 0.001$). Dots correspond to individual regression coefficients ($n = 38$ independent participants). Dots and error bars represent the trial-resolved mean ± SEM of the participant data. -: $0.05 < P < 0.1$; *: $0.01 < P < 0.05$; **: $0.001 < P < 0.01$; ***: $P < 0.001$. The brain depicted in the figure is based on a template from the software MRIcron. Chris Rorden's MRIcron, all rights reserved. https://people.cas.sc.edu/rorden/mricron/install.html.

and regions positively encoding valence (i.e., Gain > Loss). Again, we found that BOLD signal in the VMPFC jointly correlated with valence and confidence, suggesting that it plays a key role in processing a global, task-wide confidence signal (voxel-wise $P_{uncorrected} < 0.001$; cluster-wise $P_{FWE} < 0.05$; Supplementary Fig. S6c).

## Quantitative assessment of confidence-related variable encoding

Although the analysis of the qualitative patterns of activations seems to clearly point to a functional dissociation between the positive and negative prefrontal network in confidence encoding, some aspects of the demonstration still have some weaknesses. For instance, ascribing a condition-specific rather than a task-wide confidence signal to the negative network entails accepting the null hypothesis – i.e., concluding that valence and information are not statistically detectable in

the negative network ROIs' signal. Here, we propose a different set of analyses to quantitively support this conclusion without relying on this statistical caveat. To provide a fair comparison between task-wide and condition-specific confidence, we designed two new GLMs (GLM2$_{WID}$ and GLM2$_{SPE}$), that concatenated all learning contexts into one single cue-evoked event (i.e., symbol presentation period). Then, in GLM2$_{WID}$, this event was modulated by the time series of all *native* confidence judgments (i.e., the absolute confidence reports provided by our subjects on each trial). On the contrary, in GLM2$_{SPE}$, this event was modulated by the time-series of all confidence judgments, but normalized (i.e., Z-scored) per condition (i.e., reflecting variation around each condition mean). This way, the structure of these two GLMs is identical, but the parametric modulators of confidence respectively represent task-wide confidence (i.e., native, absolute confidence) and condition-specific confidence. We then extracted the confidence

regression coefficients from our ROIs, and proceeded to two types of quantitative comparisons. First, we simply compared the GLM2$_{SPE}$ and GLM2$_{WID}$ regression coefficients (Fig. 3c). In the VMPFC, activations related to native confidence were significantly higher than those related to normalized confidence ($t_{37} = 5.41$, $P < 0.001$). In total, this pattern was found in 30 out of 38 participants, further evidencing that activity in the VMPFC better corresponds to task-wide than condition-specific confidence. However, the same analysis was inconclusive for the regions of the negative network –although trending in the direction of higher activations for condition-specific confidence for some regions (DMPFC: $t_{37} = -1.40$, $P = 0.1670$; IFG + INS: $t_{37} = 0.43$, $P = 0.6684$). Note that the underlying test that was used to create ROIs, a grouping parametric effect of confidence from GLM1, was orthogonal to the follow-up tests on task-wide and condition-specific confidence encoding, therefore these analyses were not circular and did not advantage GLM2$_{SPE}$ or GLM2$_{WID}$[40]. We then complemented these analyses with a formal Bayesian model comparison (see "Methods: Bayesian model selection (fMRI)") between the GLM2$_{SPE}$ and GLM2$_{WID}$ in our ROIs, using the SPM-based MACS toolbox[41]. This time, while the analysis was inconclusive in the VMPFC (GLM2$_{WID}$ vs GLM2$_{SPE}$; Exceedance Probability EP: 48.69% vs 51.31%), it provided suggestive evidence that lateral and dorsal parts of the negative network are better explained by condition-specific than task-wide confidence (GLM2$_{WID}$ vs GLM2$_{SPE}$ EP:DMPFC: 17.78% vs 82.22%; IFG + INS: 09.66% vs 90.34%). Overall, converging evidence from different models and statistical tools seems to confirm our functional dissociation between the VMPFC and the negative network.

## Computational models for learning and confidence judgments

The vast majority of past studies investigating neurocomputational models of reinforcement learning have focused on the neural representation of learning latent variables such as option and action values, prediction errors, and various levels of (Bayesian) uncertainty. As a matter of fact, the emerging consensus in the RL literature seems to indicate that neural signal in the VMPFC is specifically linked to the representation of option values, from which decisions are derived[42,43]. Evaluating the relative merits of our current hypothesis against this consensus, namely that VMPFC encodes confidence judgments rather than values during RL, requires a computational model that faithfully captures our participants' behavior and that can produce the desired latent variables. Following the rationale of a recent study[32], we proposed a combination of a RL model and of a confidence regression, to jointly account for behavioral choices and confidence judgments exhibited in the current experimental framework (i.e., in both the learning and transfer phases). We factorially tested several families of RL model (Fig. 4a and "Methods"), which built on a basic Q-learning model (ABS), and modularly featured context-dependent learning (RELATIVE family) as well as confirmatory updating (ASYMMETRIC family)–see also refs. 39,44. Replicating previous findings, we found that both features were necessary to best account for our participant data, as revealed by a formal Bayesian Model Selection (BMS) analysis (Fig. 4b, c; winning model: RELASYM; protected Exceedance probability = 91%). The RL model provided latent variables (i.e., option Q values and context-value V), from which we then built several confidence models (Fig. 5a and "Methods"). Confidence models consisted of a logit-transformed multiple regressions that included, as predictor variables, choice difficulty–proxied by the absolute difference between option values (|Qc−Qu|)–, plus a biasing term accounting for the valence-induced bias (for which we tested several variants: 0, ΣQ, V, Qc; Fig. 5a), and an autocorrelation term (i.e., confidence in the previous trial) that accounts for the tendency of confidence judgments to exhibit serial dependency[45]. A BMS revealed that the confidence model that featured the value of the chosen option Qc as a biasing term (thereafter referred to as Qc-REG) provides the best account of participants confidence judgments (protected Exceedance probability

>99%; Fig. 5b, c). In the supplementary methods of the present paper (Supplementary Figs. S1–S5), we systematically apply the set of analyses underlying the demonstration proposed in ref. 32 and compare its results to those obtained in the present dataset (learning + transfer). This exercise confirmed that the combination of RELASYM and Qc-REG models faithfully capture our participants' behavior (choice and confidence judgments) throughout our experimental framework (learning and transfer phase), and that learning biases are fundamentally linked with confidence biases (Supplementary Fig. S5).

## BOLD activity in the positive and negative networks correlates with decision values

Thanks to the latent variables estimated from our computational models, we next tested whether activity in the prefrontal regions originally identified in our confidence analyses (Fig. 3a; VMPFC; DMPFC; IFG + INS) could also be explained with the more traditional learning and decision variables. We therefore designed a new GLM (i.e., GLM3, see Table 1) for a model-based fMRI analysis, which comprised, as parametric regressors of the cue onset, all value-related latent variables estimated by the RELASYM model: the chosen option value (Qc), the unchosen option value (Qu), and the context value (V). We then extracted the parametric regressors in the three main regions forming our confidence networks. Altogether, and in line with previous findings[34,42,43], we found that the chosen option values (Qc) correlated with BOLD activity positively in the VMPFC ($t_{37} = 3.26$, $P = 0.0023$) and negatively in the DMPFC ($t_{37} = -4.96$, $P < 0.001$) and IFG + INS ($t_{37} = -4.43$, $P < 0.001$) (Fig. 6a). In addition, the unchosen option value (Qu), correlated positively with BOLD activity in the DMPFC ($t_{37} = 2.96$, $P = 0.0053$) and IFG + INS ($t_{37} = 2.75$, $P = 0.0091$). At the whole-brain level ($P_{FWE} < 0.05$ at the cluster level), only the chosen option values (Qc) generated significant clusters of activations in the prefrontal regions, in both the VMPFC (positive) and in the IFG + INS (Supplementary Table S8). Therefore, in the context of reinforcement learning, neural activity in the ventral and dorsal prefrontal cortices can be evenly ascribed to two very different cognitive processes: the computation of decision values and/or the evaluation of confidence in the upcoming decision.

## BOLD signal in the VMPFC correlates with confidence-building variables

To evaluate whether prefrontal activations with confidence could have been purely confounded (i.e., explained) by their role in computing decision values (notably Qc, in the VMPFC), we proceeded to a reverse double-dipping exercise. We created a new GLM (i.e., GLM4), which contained the three components of confidence suggested by the Qc-REG models (Qc, |Qc−Qu|, and conf.$_{t-1}$) as parametric regressors of the cue onset. We defined as our three prefrontal ROIs the significant clusters revealed by the whole brain correlation with Qc in GLM3, and extracted the parametric regressors of the confidence component estimated with GLM4. Critically, the VMPFC ROI that was selected to be specifically associated with Qc also exhibited residual correlations with the other confidence components (Fig. 6b; Qc: $t_{37} = 3.63$, $P < 0.001$; |Qc−Qu|: $t_{37} = 1.89$, $P = 0.0657$; conf.$_{t-1}$: $t_{37} = -2.16$, $P = 0.0370$). Note, however, that when the different sources of confidence formation competed for the variance of BOLD signals, only Qc elicits whole-brain significant activations in VMPFC (voxel-wise $P_{uncorrected} < 0.001$; cluster-wise $P_{FWE} < 0.05$; Supplementary Table S9). Still, those analyses suggest that the VMPFC does not simply encode Qc, but exhibit additional signatures of confidence signal. Though regions of the negative network (DMPFC and IFG + INS) also seem to correlate with additional confidence variables, above and beyond Qc (notably and most robustly conf.$_{t-1}$; DMPFC: $t_{37} = -3.46$, $P = 0.0014$ and IFG + INS: $t_{37} = -4.32$, $P < 0.001$; Fig. 6b), the set of analyses dissociating Qc from confidence encoding seems less relevant there. Indeed, the negative network has been less systematically associated with value encoding in

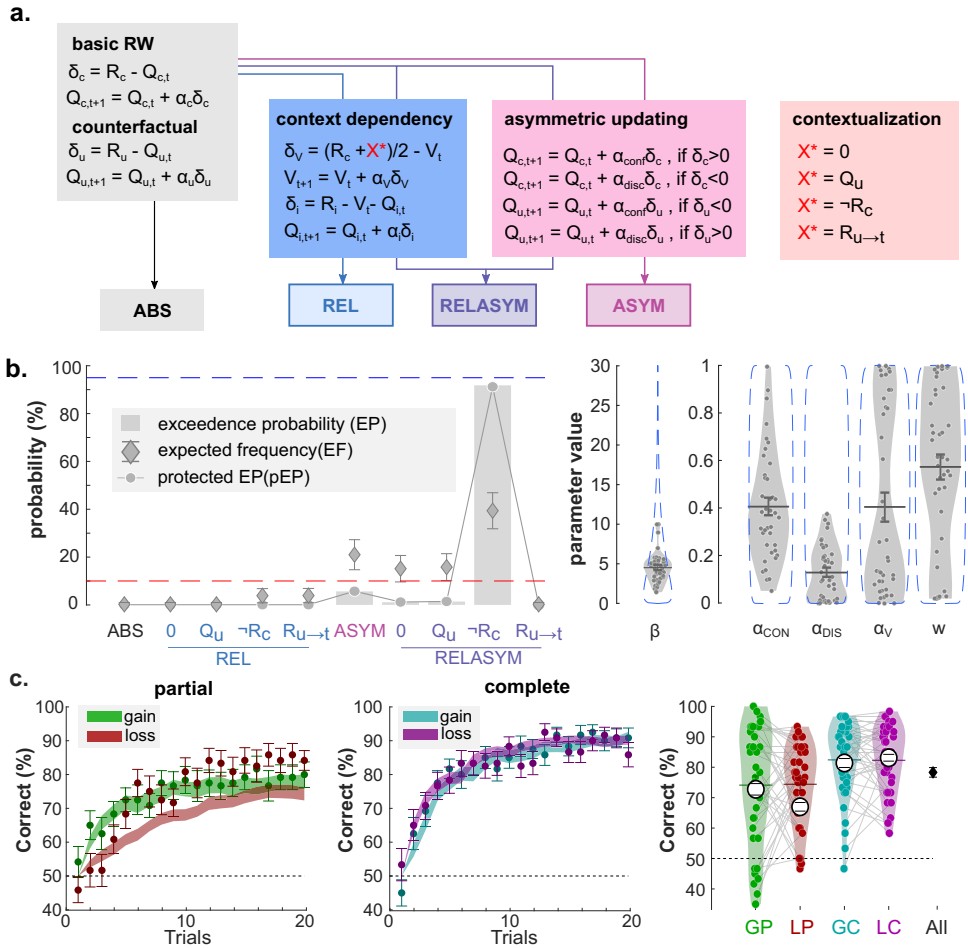

**Fig. 4 | Modeling choices in the learning phase. a** The learning model architecture. Color panels represent different components of value updating rules. Gray panel: Absolute model (ABS), which consists of basic delta update rule. Blue panel: Relative model (REL); Pink panel: Asymmetric updating model (ASYM); Purple panel: relative-asymmetric model (RELASYM). The contextualization panel illustrate how the unchosen option (X*) is updated in the partial information condition, when the unchosen option outcome (Ru) is not available. X* can take the value of 0, of the expected unchosen value (Qu), of the paired outcome (¬R) and of the last seen outcome associated with the option (Ru→t). **b** Left panels: Bayesian model comparison Between models included in the model space (X-axis). The Y axis indexes the value of three BMC criteria, namely exceedence probability (EP; gray histograms), expected frequencies (EF; diamonds) and protected exceedance probability (pEP; line and dots) of each model. The red dashed line represents the guessing level for EF. The blue dashed line represents the threshold (95%) for the exceedance probability. Right panels: Estimated parameter values of the winning model (RELASYM, X* = with ¬Rc). Dots represent individual data points ($n = 40$ independent participants). Error bars displayed within the violin plots indicate the

sample mean ± SEM. The blue, dotted envelop represent the prior distribution. **c** Left: modeled trial-by-trial percentage of correct responses. Dots and error bars represent the mean ± SEM of the participant data. Filled, shaded colored areas represent mean ± SEM of the posterior predictive fits obtained from our winning computational model (RELASYM, X* = with ¬Rc). Right: average percentage of correct responses across conditions at the individual level (colored dots; $n = 40$ independent participants) and group-level (horizontal bars). The black error bars indicate the overall performance over conditions. The colored horizontal bar and error bar represent the mean and SEM of the real data, respectively. The large white dot and corresponding error bar represent mean ± SEM of the posterior predictive fits obtained from our winning computational model (RELASYM, X* = with ¬Rc). $Q_{c/u,t}$: value of the chosen/unchosen option at trial t. $R_{c/u}$: outcome associated to the chosen/unchosen option. $\delta_{c/u}$: prediction error for the chosen/unchosen option. $\alpha_{u/c}$: learning rate for the chosen/unchosen option. $\alpha_{conf/disc}$: learning rate for confirmatory/disconfirmatory information. $V_t$: context value; $\delta_V$: prediction error for the context value. $\alpha_V$: learning rate for the context value.

previous studies. In addition, analyses reported in Fig. 6a already show that signal in these regions not only correlates with the value of the chosen option (Qc), but also robustly (with an opposite sign) with the value of the unchosen option (Qu). This pattern is tentatively consistent with a role in the comparison of available options (rather than valuation), including potentially the context-specific confidence associated with this comparison.

**BOLD signal in the VMPFC is better explained by confidence than decision variables**
A recent stream of studies has suggested that, in simple decision-making or judgment situations, the VMPFC encodes a combination of both decision values and confidence[3,27,46,47]. In this last section, to

refine the characterization of VMPFC activity during human reinforcement learning, we estimated an fMRI model which included both Qc and confidence judgments as parametric regressors (GLM5). Following the rationale of ref. 3, we designed value-related VMPFC ROIs, from the Qc-activations revealed in GLM3, and from a meta-analysis of fMRI activations value[48]. We then extracted regression coefficients of Qc and confidence from the GLM5 model, so as to test for the presence of confidence signals in those value-coding regions (Fig. 7a). Despite the choice of our ROIs, which should bias our analyses in favor of value activations, the Qc-related activations were marginal to insignificant (Fig. 7a, GLM3-ROI: $P = 0.0553$; Bartra ROI: $P = 0.2324$) in our model in which value- and confidence-related parametric modulators compete for variance. On the contrary, confidence-related activations were

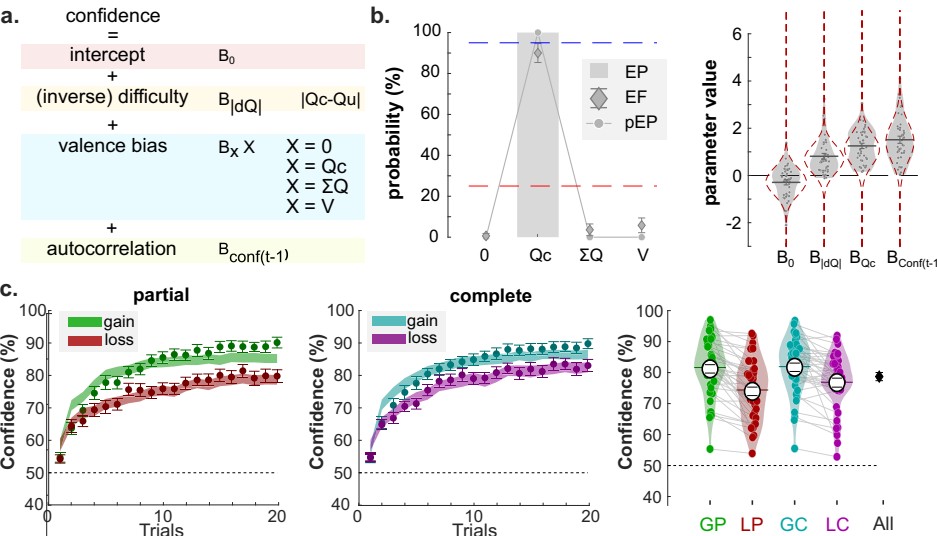

**Fig. 5 | Modeling confidence in the learning phase. a** The confidence model architecture to explain participants' confidence judgment data. Color panels represent different components of the multiple regression predicting confidence. In particular, the blue rectangle pictures the different hypotheses for the biasing term. **b** Left panels: Bayesian model comparison. *X* axis represents the models with different hypothesized valence biases. Y-axis represents the value of three criteria, including exceedance probability (EP; gray histograms), expected frequencies (EF; diamonds) and protected exceedance probability (pEP; line and dots) of each model. The red dashed line represents the guessing level for EF. The blue dashed line represents the threshold (95%) for the exceedance probability. Right panels: Estimated parameter values of the winning model (Qc-REG). Dots represent individual data points. Error bars displayed within the violin plots indicate the sample mean ± SEM. The blue, dotted envelop represent the prior distribution. **c** Left: modeled trial-by-trial confidence judgments. Dots and error bars represent the mean ± SEM of the participant data (*n* = 40 independent participants). Filled, shaded colored areas represent mean ± SEM of the posterior predictive fits obtained from our winning model (Qc-REG). Right: average confidence across conditions at the individual level (colored dots) and group-level (horizontal bars). The black error bars indicate the overall performance average across conditions. The colored horizontal bar and error bar represent the mean and SEM of the read data, respectively. The large white dots and corresponding error bar represent mean ± SEM of the posterior predictive fits obtained from our winning computational model (Qc-REG). $Q_{c/u}$ value of the chosen/unchosen option, *V* context value, $\Sigma Q$ sum of chosen and unchosen *Q* values.

clearly significant in both ROIs (Fig. 7, *P*s < 0.001), and significantly larger than Qc-related activations (Fig. 7a, *P*s < 0.05). Note that a formal comparison between models featuring one (Qc or confidence) versus two (Qc and confidence) using BMC failed to provide conclusive results. In the negative network (DMPFC; INS + IFG), the comparison of confidence and Qc-parametric regressors did not reach significance, again suggesting a functional dissociation with its positive counterpart (Supplementary Fig. S7).

Finally, we considered the possibility that value and confidence signals dominate in different sub-regions of the prefrontal cortex[49]. Therefore, following the rationale in refs. 29,49, instead of averaging signal over the entire ROI, we extracted regression coefficients in a large anatomical prefrontal ROI, and marginalized those activations along the anterio-posterior (Y) and ventro-dorsal (Z) axes (Fig. 7b). This finer-grained analysis revealed that confidence activations dominate value-activations over all portions of the medial prefrontal cortex.

## Discussion

Decisions are usually accompanied by confidence judgments, which reflect subjective (un)certainty about the choice being correct[2–4]. This internal signal plays a crucial role in guiding behaviors and has been associated with two main prefrontal networks: VMPFC and DMPFC[18,19,24]. To date, though, the relative contribution of those two networks in the mechanisms underlying confidence formation remains unclear. To fill this gap, we combined fMRI and an adapted probabilistic reinforcement learning task[31,33,34], in which we systematically manipulated two dimensions of the learning context: the valence of the outcome (gain vs. loss) and the outcome information (partial vs. complete feedback). At the behavioral level, we successfully replicated the valence effect on confidence judgments: confidence is significantly higher when learning to gain rewards relative to learning to avoid

losses, despite participants learning equally well in both contexts[31–33]. At the neural level, we first replicated consensual and established results: confidence was positively related to the activation in the VMPFC and neighboring area pgACC (positive-confidence network) and negatively related to the activation in the DMPFC, IFG, and INS (negative-confidence network)[18,19,24]. Then, we uncovered two key findings. First, our analyses revealed that VMPFC activity represents a task-wide (subjective) confidence signal as it tracks confidence within contexts together with the valence bias that increases confidence in gain contexts. Activation in the negative-confidence network (DLPFC, DMPFC), on the other hand, only tracks condition-specific confidence. Accordingly, we speculated that the VMPFC is a key region involved in the valence-induced confidence bias during reinforcement learning. Second, we found that, contrary to the current dominant view in the field, the activation in the VMPFC can be better explained by confidence rather than other value-related variables estimated by a RL model. In the following sections, we discuss these findings in more detail.

The simultaneous neural representation of valence and confidence in the VMPFC suggests that VMPFC integrates affective/motivational information with metacognition, and as such plays a key role in the valence-induced confidence bias[3,27,29,46,50–53]. Contrary to our theoretical predictions, we did not identify a brain region that is sensitive to confidence and to the information manipulation (i.e., partial and complete feedback). This might be due to the low effect size of information on confidence (though effects on accuracy are clear) or the fact that, as our modeling suggests, participants tend to infer the counterfactual outcome when not observed—see Fig. 4 and refs. 32,54. Another possibility is that, while confidence-related variables are explicitly monitored by some brain areas, uncertainty is implicitly encoded in the variance of neural populations, which our current neuroimaging approach would fail to capture[55,56].

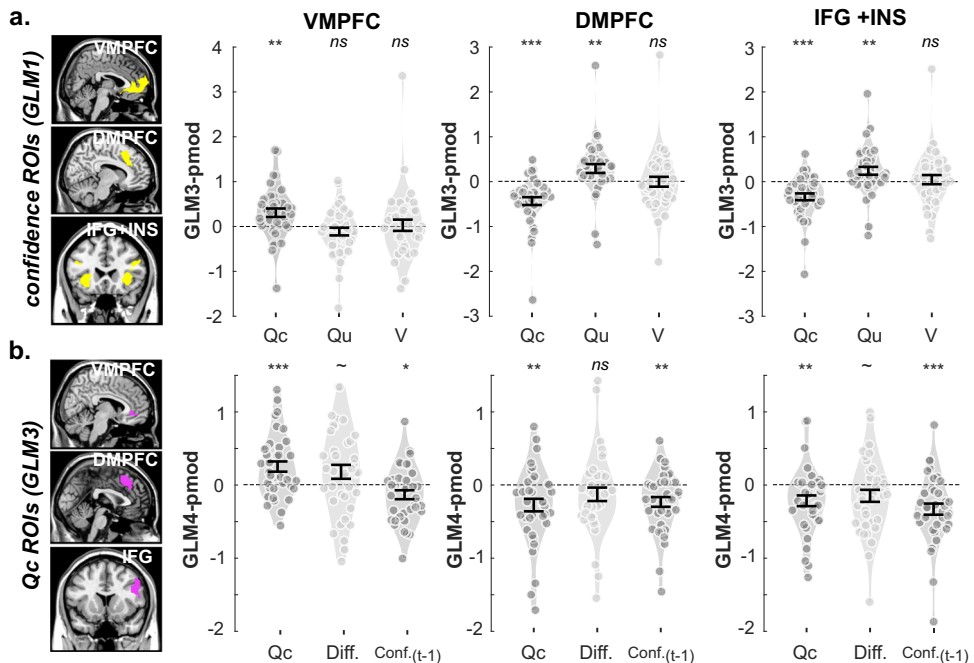

**Fig. 6 | vmPFC is involved in value and confidence processing.** Violin plots represent the sample distribution of fMRI regression coefficients corresponding to several variables of interest included in different GLMs, extracted from each ROI (left: VMPFC; middle: DMPFC; right: IFG + INS) at the symbol presentation phase ($n = 38$ independent participants). **a** Regression coefficients for RL-derived value latent variable. Dots correspond to individual regression coefficients. One-sample $t$ tests indicated that all regions significantly encode the chosen $Q$ value (two-sided tests; VMPFC: $t_{37} = 3.26$, $P = 0.0023$; DMPFC: $t_{37} = -4.96$, $P < 0.001$; IFG + INS: $t_{37} = -4.43$, $P < 0.001$), and regions of the negative network additionally encode the unchosen option value Qu (DMPFC: $t_{37} = 2.96$, $P = 0.0053$; IFG + INS: $t_{37} = 2.75$, $P = 0.0091$). **b** Regression coefficients for confidence model latent variables. One-sample t-tests indicated that the VMPFC ROI that was selected to be specifically associated with Qc also exhibited residual correlations with the other confidence

components (two-sided tests; Qc: $t_{37} = 3.63$, $P < 0.001$; Diff: $t_{37} = 1.89$, $P = 0.0657$; conf.$_{(t-1)}$: $t_{37} = -2.16$, $P = 0.0370$). ROIs were defined using the confidence contrast from GLM1 (**a**) or the Qc-contrast from GLM3 (**b**), with standard significance thresholds (one-sided tests; $p_{uncorrect} < 0.001$, cluster-wise $P_{FWE} < 0.05$). Dots correspond to individual regression coefficients. Dark gray and light gray indicate the effect is significantly and insignificantly different from 0, respectively. Error bars represent mean ± SEM. Qc: parametric modulator of chosen option; Qu: parametric modulator of chosen option.; V: parametric modulator of context value.; Diff.: parametric modulator of absolute value difference of Qc and Qu. -: $0.05 < P < 0.1$; *: $0.01 < P < 0.05$; **: $0.001 < P < 0.01$; ***: $P < 0.001$. The brain depicted in the figure is based on a template from the software MRIcron. Chris Rorden's MRIcron, all rights reserved. https://people.cas.sc.edu/rorden/mricron/install.html.

In addition, our results provide evidence for the co-existence of task-wide confidence in VMPFC and condition-specific confidence in DMPFC. This functional difference confirms that those two brain networks are not redundant in the way they process confidence-relevant information[18,24], but also raise legitimate questions about the advantages of tracking both variables and the relation between them. Naturally, access to task-wide (i.e., absolute) confidence is critical to compare (or even choose between) different choice situations whose assessment regarding the probability of being correct differ[57]. Task-wide confidence can be viewed as an overarching estimate of confidence that enables to select situations in which we perform well, and avoid situations in which we perform less well. Its role of monitoring confidence across multiple contexts therefore places task-wide confidence in an advantageous position to solve the explore-exploit dilemma. Yet, evidence suggest that most neural and cognitive computations are context-dependent[58,59], notably in the context of reinforcement learning[39,60], such that metacognition and confidence might not elude this general neurocomputational principle. While our current results remain agnostic about the mechanistic interactions between task-wide and condition-specific confidence, most models of confidence formation seem to assume that local variables (e.g., uncertainty or condition-specific confidence) are precursors of more general, absolute confidence judgments[4,16,61]. In our case, this would imply that early, condition-specific signals in the negative network (DMPFC, DLPFC) are then fed to the positive network (VMPFC), where a general, task-wide confidence signal matches

the report of participants which corresponds to the subjective experience—i.e., phenomenological dimension—of the feeling of confidence[24,62]—but see ref. 28 for evidence of opposite patterns. Finally, a couple of recent studies investigated how a global feeling of confidence (over a whole task) builds from multiple local signals (over trial-by-trial changes in task difficulty and performance), and report that VMPFC tracks local confidence in a manner that is sensitive to global self-performance estimations[61,63]. Altogether, these results seem to indicate that VMPFC aggregates complex confidence estimates over multiple layers of precursor variables.

Two main lines of arguments motivated us to complement our first set of neuroimaging analyses focused on confidence signals with model-based assessments of value-related signals. First, similarly to the decision-making literature, the reinforcement-learning literature has so far mostly associated VMPFC with the processing of value—rather than confidence[42,43]. Second, we recently suggested that, during reinforcement-learning, confidence builds notably on two variables estimated from learned option-values: the choice difficulty (proxied by the absolute difference between the two available options values), and the chosen option value (Qc)[32]. This leaves open the possibility that the activations that we originally associated with confidence in VMPFC actually encode the *sources* of confidence (i.e., value signals) rather than confidence per se. To address these concerns, we used the same modeling strategy proposed in ref. 32, and first confirmed their conclusions regarding both learning and confidence models. Indeed, our results showed that the participants choice behavior can be best

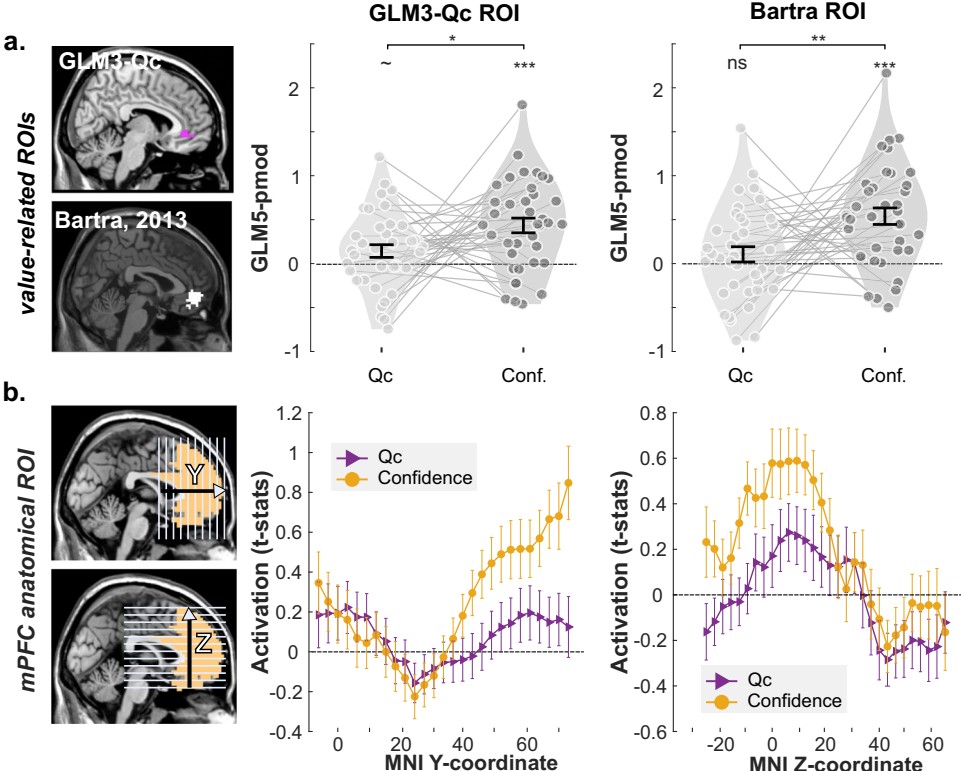

**Fig. 7 | value and confidence activations in the VMPFC. a** ROI analysis with Qc-related ROIs identified in the present study (top-left; purple areas) and in an independent meta-analysis (bottom-left; white area). Right: the regression coefficients corresponding to Qc and confidence in GLM5 were summarized at the individual level (dots). Violin plots represent the sample distribution of fMRI regression coefficients. Dots correspond to individual regression coefficients ($n = 38$ independent participants). Paired $t$ tests indicated that VMPFC consistently better encode Confidence than Qc (two-sided tests; GLM3-Qc ROI: $t_{37} = -2.13$, $P = 0.0399$; Bartra ROI: $t_{37} = -2.85$, $P = 0.0070$). **b** Anatomical ROI of mFPC. BOLD signal was extracted along y-dimension from posterior to anterior area and along z-dimension from ventral to dorsal area (pictured slices are only illustrative, and do not indicate the actual coordinate of the extracted signal). Voxel-wise $t$ values of Qc and confidence in GLM5 were extracted and averaged over two dimensions. Middle: average $t$ value along MNI y-coordinate. Right: average $t$ value along MNI y-coordinate. Dots and error bars represent mean ± SEM. Qc: parametric modulator of chosen option; Conf.: parametric modulator of confidence ratings. -: $0.05 < P < 0.1$; *: $0.01 < P < 0.05$; **: $0.001 < P < 0.01$; ***: $P < 0.001$. The brain depicted in the figure is based on a template from the software MRIcron. Chris Rorden's MRIcron, all rights reserved. https://people.cas.sc.edu/rorden/mricron/install.html.

explained by a reinforcement-learning model featuring context-dependent learning, and confirmatory updating (Supplementary Fig. S1). Additionally, we did confirm that confidence judgments are best explained by a linear combination of choice difficulty (proxied by the absolute difference between the two available option values) and the chosen option value (Qc) as a biasing term—akin to a choice-congruent evidence integration bias[64–66]. This model provides an excellent fit to participants' choices and confidence judgments in both learning and transfer phases, and generates key behavioral patterns observed in our data, suggesting that it adequately tracks the cognitive operations mobilized to solve our task. Thereby, the model-derived latent variables allow us to investigate the neural correlate of valuation during learning[67]. Note that contrary to most previous studies, our design allowed the separation of option evaluation and motor mapping, which minimizes the potential action-related effect on the correlation between BOLD signal and decision-related variables such as values and confidence[68]. In this context, we confirmed that the value of the chosen option correlates positively with BOLD signal in the VMPFC[69–72]. More dorsal and lateral regions of the prefrontal cortex (DMPFC, DLPFC) appear to encode with opposite signs the value of the chosen and unchosen options. This pattern could be consistent with the idea that value comparison is effectuated in these more dorsal prefrontal regions[73–76], and could provide an estimate of the value of control or of information[77,78].

To bridge these results on valuation in vmPFC with results suggesting confidence encoding in the same region, we investigated whether the VMPFC encodes additional confidence precursors (e.g., choice difficulty) in addition to Qc. ROI analyses revealed significant correlations between the activation in the VMPFC and all three confidence precursors identified by our confidence model, suggesting that VMPFC does not simply encode Qc. Consolidating these results, we also found that the activation in the VMPFC can be better explained by confidence than Qc when both variables are included in a single model, and this is observed regardless of the level of granularity considered. Note that, to avoid the double-dipping issue, we selected ROIs that are related to chosen option value from the present study and an independent literature[48], therefore favoring de-facto the opposite hypothesis, namely that VMPFC would preferentially encode Qc. The fact that confidence signals dominate value signals in the VMPFC clashes with the current understanding of its functional role in reinforcement-learning task, which is almost exclusively restricted to option valuation and representation of cognitive maps[77].

There are at least three tentative explanations for this apparent discrepancy. First, our results could be compatible with the idea that VMPFC does uniquely encode Qc (rather than confidence), but this latent variable is not well estimated by the RL model to robustly capture VMPFC signal variance. In our present modeling exercise as well as a previous modeling paper[32], we tried to nullify this possibility by going to great length to show that our RL and confidence models can qualitatively and quantitatively account for choice behavior and confidence judgment (Supplementary Fig. S3). Interestingly, in the (possible) case that a misfit persists and that the Qc variable is mis-

estimated, our present results suggest that eliciting confidence judgments could help researchers to better identify the neural networks engaged in value-based learning. Second, similarly to what has recently been shown in decision-making, VMPFC might actually jointly represent decision values and confidence during reinforcement-learning[3,27,47]. In our data, only small portions of the VMPFC (anterior and ventral) still correlate positively with Qc when confidence is included in the model. Finally, it is possible that the presence of confidence elicitation in the present study somewhat affects the other computations related to valuation and decision. Although previous work suggests that value and confidence encoding in the VMPFC are both automatic[3,47,79,80], an increasing number of studies also reported that VMPFC (value) coding depends on incidental emotional states, as well as specific goals and demands of the task at hand[16,81,82]. These last two possibilities are consistent with the idea that the role of medial and orbital frontal cortex in decision-making and flexible behavior is more complex than initially thought, and might deserve further (re) investigations[77,83]. A recent study even suggests that, in a task where participants must form beliefs about the accuracy of reward information cues by trial-and-error, the polarity of uncertainty (i.e., inverse confidence) encoding in the VMPFC could reverse, from positive during exploration to negative during exploitation[84]. In our taxonomy, that would mean that VMPFC can be first part of the negative network (because our reference is confidence rather than uncertainty) and can then gradually switch to being part of the positive network. Further research should identify under which conditions the polarity of confidence signals in the VMPFC could possibly change.

In the present study, confidence is non-instrumental, and only consists in a read-out of the subjective choice accuracy. In numerous ecological contexts, confidence can be key to monitor and adapt behavioral strategies. Given the multiple layers of confidence and uncertainties uncovered here and the functional dissociations of their neural underpinnings, future studies will need to consider which variable (objective uncertainty, condition-specific confidence, task-wide confidence) and which (confidence) biases impact future behavior—and how. This last point is critical for developing interventions targeting confidence biases, especially as confidence dysfunctions are increasingly seen as relevant markers in clinical applications[85,86].

## Methods

### Participants

Forty participants (female = 23; Age = 22.69 ± 4.44) were recruited from the subject pool of the behavioral science lab (https://www.lab.uva.nl/lab) and through poster adverts distributed on the University of Amsterdam (UvA) campus. The ethical approval was obtained from the Faculty Ethics Review Board (FMG-UvA) at UvA (reference number: 2018-EXT-9205). Before the experiment, only participants that passed the prescreening procedure (e.g., no claustrophobia, no metal in the body) were invited to come to the MRI scanner and were sent an invitation email and detailed information about the experiment and MRI. Participants were asked to arrive at the laboratory 30-min before the experiment. Once participants arrived, they gave informed consent and read the instruction again. Afterward, they experienced a 16-trial practice with the same learning task (but using different symbols) as well as a lottery incentivize procedure outside of the MRI scanner.

The final payout was computed as follows: show-up fee (20€), accumulated outcome from the learning task and bonus from the confidence incentivization procedure. The mean and standard deviation of the payout was 32.18 ± 3.46€. All the tasks were implemented using MatlabR2015a® (MathWorks) and the COGENT toolbox.

### Probabilistic instrumental-learning tasks

We adopted our previous instrumental reinforcement learning task[31,33,34] for fMRI by adding incentivized confidence ratings and by separating symbol evaluation and motor response in each trial (see details below). Participants were asked to maximize payoff during the learning task by choosing the symbol with the higher expected value in a pair at each trial (Fig. 1). In each run of 80 trials, four fixed pairs of abstract symbols were used to represent four conditions in the two (feedback valence: gain or loss) by two (information: partial or complete) within-subjects design (Fig. 1b). Specifically, eight symbols were divided into four fixed combinations and are constantly arranged to gain & partial (GP), loss & partial (LP), gain & complete (GC), and loss & complete (LC) conditions. Each pair of symbols indicated a specific condition and possible outcomes. For example, for gain contexts (i.e., GP and GC), the possible outcomes are +€1 or +€0.1. Conversely, −€1 or −€0.1 are possible outcomes in the loss contexts (i.e., LP and LC). The probabilistic outcome of an option was determined by reciprocal but independent probabilities, 75% or 25% (Fig. 1b). The symbol that enjoys a higher expected value ($\sum$ probability × outcome) was defined as the correct option in each pair. Note that only the chosen outcome was added to the final payoff in both the incomplete and complete feedback conditions.

All the participants completed three runs of 80 trials, such that each of the four conditions (i.e., each pair of symbols) was repeated 20 times per run. In each trial (Fig. 1a), the symbols were presented first (1500–3500 ms; mean = 2050 ms). To avoid the potentially confounding influence of motor responses during symbol evaluation, the symbols disappeared for a while (500–3000 ms; mean = 800 ms) after symbol presentation. Afterward, two white bars appeared on either right or left of the location of the invisible symbol to indicate which button should be pressed to select the corresponding symbol (i.e., the right button: the white bar was on the right side of the symbol). Once a decision was made, two red bars were displayed beside the chosen symbol (500 ms). Before seeing the outcome, participants were asked to state their confidence about choosing the symbol that is better on average (i.e., with a higher expected value). Confidence ratings were done on a scale ranging from 50% to 100% with incremental steps of 5%, and randomized starting points and without time constraints. At the end of each trial, participants were shown the outcome from the chosen option only in the partial information conditions (i.e., GP and LP) for 2000 ms. Otherwise, both chosen and unchosen outcomes were displayed in the complete information conditions (i.e., GC and LC. See Fig. 1b).

In order to motivate participants to accurately report confidence, confidence judgments were incentivized by a Matching Probabilities (MP) mechanism, a well-validated method from behavioral economics adapted from the Becker-DeGroot-Marschak auction[87,88]. Specifically, we randomly selected three trials from three runs (i.e., one trial/ run) and then compared the confidence rating $p$ at that trial with a random number $r$ (chosen from the range between 50% and 100%). If $p \geq r$, then participants won the bonus of 5€ when the chosen symbol indeed had the higher expected value (i.e., the correct one), otherwise, participants won nothing. If $p < r$, participants won the bonus of 5€ with a probability of $r$, otherwise, won nothing with a probability of $1 − r$. The euros earned from the game were exchanged for the actual money with a certain exchange rate (1 EU in game = 0.3 payouts EU). Again, all participants were informed about the rule of payout and experienced practice trials in both the learning task and confidence incentivization before the real experiment in the MRI scanner.

### Transfer task

After the learning task, participants left the scanner and were instructed to perform an additional transfer task, where each symbol from the last run of the learning task was paired with all other seven symbols (i.e., forming 24 new and 4 original pairs). Participants were asked to choose one symbol that can benefit them more, and rate their confidence in their choice. No feedback and monetary incentives were offered in this task. However, participants were asked to imagine that they were able to earn money from the chosen symbols. Because the

present study focuses on the neuroimaging data, which was only available for the learning task, analyses of choices from the transfer task are not detailed in the Main Text (but see Supplementary Figs. S2, S4, and S5).

## Behavioral analyses

In this study, we mostly focused our analyses on three dependent variables of interest available during the learning task: choice accuracy, reaction times, and confidence. The choice accuracy referred to the probability of choosing the relatively better symbol in a pair of symbols (i.e., the one with a higher expected value). The reaction time was defined as the time between the onset of the cues allowing response (referred to as the choice screen in Fig. 1a) and the actual (self-paced) choice. Confidence simply corresponded to the rating elicited in the confidence judgment screen. To test for the effect of valence and information manipulations, as well as their interaction, these measures were averaged over three runs for each condition and participants and were then fed into two-way repeated-measures ANOVAs. The direction of changes was analyzed by follow-up t-tests. In particular, one-sample t-tests were used when comparing data to a reference value (e.g., guessing level: 50%), and paired t-tests were used to compare responses across different conditions (e.g., gain vs. loss) and different measures (e.g., averaged learning performance vs. averaged confidence).

All statistical analyses were performed using MatlabR2021a® (MathWorks) and its built-in functions (i.e., one-sample t-test: t-test; paired t-test: ttest2; repeated ANOVA: anovan; Pearson's correlation: corr), with a statistical significance level of alpha 0.05. Unless otherwise specified, significance level for t-tests correspond to two-tailed hypothesis test.

## Computational modeling−methods

**Learning models—structure and model space.** Participants' choices from both learning task and transfer task were fitted with ten reinforcement learning models (RL models) proposed in ref. 32. The models in the model space can be categorized into four families: ABSOLUTE model (ABS), RELATIVE models (REL), ASYMMETRIC models (ASYM), and RELATIVE-ASYMMETRIC models (RELASYM).

The ABS model is the baseline model. Other models were built up based on the ABS model and assumed other sources of information were integrated during learning (Fig. 4a).

In the ABS model, in all learning contexts $s$, both chosen option value $Q(s,c)$ and unchosen option value $Q(s,u)$ are updated through a delta-rule function at trials $t$:

$$Q_{t+1}(s,c) = Q_t(s,c) + \alpha_c \times \delta(s,c)$$
$$Q_{t+1}(s,u) = Q_t(s,u) + \alpha_u \times \delta(s,u) \qquad (1)$$

where $\alpha_c$ and $\alpha_u$ are learning rates and $\delta$ referred to the prediction error. The prediction error is defined as the difference between the estimated option value Q and the real outcome R:

$$\delta_c = R_t(s,c) - Q_t(s,c)$$
$$\delta_u = R_t(s,u) - Q_t(s,u) \qquad (2)$$

The RELATIVE and RELATIVE-ASYMMETRIC families of models feature context-dependent learning[34,39,89]. Thereby, the prediction errors for chosen and unchosen options are corrected with the context value $V(s)$ as follows:

$$\delta_c = R_t(s,c) - V_t(s) - Q_t(s,c)$$
$$\delta_u = R_t(s,u) - V_t(s) - Q_t(s,u) \qquad (3)$$

where the context value is also updated through delta-rule with its own learning rate $\alpha_V$ and prediction error $\delta(s,v)$:

$$V_{t+1}(s) = V_t(s) + \alpha_V \ \delta(s,v) \qquad (4)$$

When the counterfactual outcome is available (i.e., complete information conditions), the prediction error for context value is computed as the difference between the estimated context value and the average outcome values:

$$\delta_V = (R_t(s,c) + R_t(s,u))/2 - V_t(s) \qquad (5)$$

When the outcome for the unchosen option was not available in context s (i.e., partial information conditions), we assume participants infer an approximation of it $X^*$, and calculated the prediction error for context value accordingly:

$$\delta_V = (R_t(s,c) + X^*)/2 - V_t(s) \qquad (6)$$

We tested four alternatives for this approximated inference $X^*$, which were implemented in different models. These four alternatives are 0, unchosen option value ($Q_t(s,u)$), the last experienced unchosen outcome for the unchosen option ($R_{t-1}(s,u)$), and weighted *imaginary forgone outcome* ($w \times \neg R_t(s)$). Following on our previous work[32,54], the imaginary forgone outcome is determined by the sign of context value ($V_t$) and the magnitude of the received outcome ($R_t(s,c)$):

$$\neg R_t(s) = \begin{cases} 1 \ if \ |R_t(s,c)| = 0.1 \ and \ V_t(s) > 0 \\ -1 \ if \ |R_t(s,c)| = 0.1 \ and \ V_t(s) < 0 \\ 0.1 \ if \ |R_t(s,c)| = 1 \ and \ V_t(s) > 0 \\ -0.1 \ if \ |R_t(s,c)| = 1 \ and \ V_t(s) < 0 \\ 0 \ if \ V_t(s) = 0 \end{cases} \qquad (7)$$

$\neg R_t$ is multiplied by a weight parameter $w$ ($0 \le w \le 1$).

The ASYMMETRIC and RELATIVE-ASYMMETRIC families of models feature asymmetric updating. This follows from previous studies, that demonstrated the presence of a choice-confirmation bias in reinforcement-learning contexts[44,90,91]. The models capture this bias by allowing two different learning rates (i.e., $\alpha_{CON}$ and $\alpha_{DIS}$) to weight the prediction-error in the value-updating process, depending on the sign of the prediction error. In particular, $\alpha_{CON}$ (confirmatory learning rate) weights the positive prediction error for chosen option and the negative prediction error for unchosen options. By contrast, $\alpha_{DIS}$ (disconfirmatory learning rate) weights the negative prediction error for chosen options and the positive prediction error for unchosen options.

$$Chosen \ option \begin{cases} Q_{t+1}(s,c) = Q_t(s,c) + \alpha_{CON} \times \delta(s,c), if \ \delta(s,c) > 0 \\ Q_{t+1}(s,c) = Q_t(s,c) + \alpha_{DIS} \times \delta(s,c), if \ \delta(s,c) < 0 \end{cases} \qquad (8)$$

$$Unchosen \ option \begin{cases} Q_{t+1}(s,u) = Q_t(s,u) + \alpha_{CON} \times \delta(s,u), if \ \delta(s,u) < 0 \\ Q_{t+1}(s,u) = Q_t(s,u) + \alpha_{DIS} \times \delta(s,u), if \ \delta(s,u) > 0 \end{cases} \qquad (9)$$

Finally, choice probability between two options (A, B) of the same context s in the learning task is computed with the softmax function:

$$P_{learning}(s,A) = (1 + \exp(\beta(Q_t(s,A) - (Q_t(s,B)))))^{-1} \qquad (10)$$

The same softmax function and the same inverse temperature parameter β are applied to model choices in the transfer task between two given options C and D belonging to learning contexts $s_C$ and $s_D$:

$$P_{transfer}(s_C,s_D,C) = (1 + \exp(\beta(Q_{end}(s_C,C) - (Q_{end}(s_D,D)))))^{-1} \qquad (11)$$

where $Q_{end}(s_C,C)$ and $Q_{end}(s_D,D)$ are the $Q$ values for options C and D estimated at the end of the learning task in their respective learning contexts.

**Learning models—model optimization and comparison.** Parameter optimization was performed by minimizing the negative logarithm of the posterior probability (n*LPP*)[92]:

$$nLPP = -\log(P(\theta_M|D,M)) \propto -\log(P(D|M,\theta_M)) - \log(P(\theta_M|M)) \quad (12)$$

$P(D|M,\theta_M)$ refers to the likelihood of the observed data $D$ (i.e., sequence of choices) given the current model M and its parameters $\theta_M$. $P(\theta_M|M)$ refers to the prior probability of the parameters.

We used broad priors based on the literature[93]: The prior distributions of learning rates ($\alpha$) and imaginary outcome weight ($w$) were defined as beta distributions (Beta(1.1, 1.1) in MATLAB), and the prior distribution of the inverse temperature parameter $\beta$ was defined as a gamma distribution (Gamma(1.2, 5) in MATLAB). Parameter search was initialized from random starting points selected from certain ranges (i.e., $0 < \alpha < 1$; $0 < w < 1$ ; $0 < \beta < \infty$) and used an L-BFGS-B algorithm implemented via Matlab's *fmincon* function[94].

For model comparison, we calculated, for each individual, the Laplace approximation to the model evidence (LAME), which penalizes model complexity (i.e., number of parameters) as follows:

$$LAME = -nLPP + \frac{df}{2}\log(2\pi) - \frac{1}{2}\log|H| \quad (13)$$

where $n$ is the number of trials, $df$ is the number of free parameters and $H$ is the Hessian.

Quantitative model comparison was performed via a formal BMS random-effect procedure[95] and implemented in the mbb-vb-toolbox (http://mbb-team.github.io/VBA-toolbox/). This toolbox performs the Bayesian model selection procedure and estimates two indicators: the expected frequencies (EF) and the exceedance probability (EP) for each model. Specifically, the expected frequency *EF* of a model quantifies the probability that the model generated the data for any randomly selected subject. Note that the EF should be higher than chance level given number of models in the model space. EP, on the other hand, quantified the belief that the model is more likely than all the other models of the model-space.

Note that parameter recovery and model recovery for the learning models are detailed in ref. 32.

**Confidence models—structure and model space.** Participants' confidence ratings were separately fitted in the learning task and in the transfer task with four confidence models proposed in[32]. Confidence models are defined as logit-transformed multiple linear regression models that use the latent variables estimated by the winning RL model (i.e., RELASYM) to predict confidence ratings (Fig. 5a). Each model consists of one intercept and two predictors: (1) task difficulty, which is measured as absolute value difference between options (|Qc-Qu|) and (2) a hypothesized source of valence bias. We tested four hypothesized sources of valence bias: none (0), the summed value of available options ($\Sigma Q = Qc + Qu$), the expected value of the chosen option (Qc), and the context value (V). In the learning task, this latter was straightforwardly available as $V_t(s)$. In the transfer task, we generalized the idea of context value for choice between any two options C and D, as $V = \frac{V_{end}(s_C) + V_{end}(s_D)}{2}$, where $V_{end}(s_C)$ and $V_{end}(s_D)$ are the (choice-independent) values associated with the original contexts of options C and D estimated at the end of the learning task. In addition to these two predictors, the models for the learning task contains an additional predictor capturing the fact that confidence in the current trial is usually influenced by confidence in the previous trial: an autocorrelation term $Conf_{t-1}$. Ultimately, confidence models can be expressed as followed:

$$Learning\,task: y_t = \varphi\left(B_0 + B_{|dQ|} \cdot \Delta Q_t + B_x \cdot bias_t + B_{conf(t-1)} \cdot y_{t-1} + \epsilon\right), \quad (14)$$

$$Transfer\,task: y_t = \varphi(B_0 + B_{|dQ|} \cdot \Delta Q_t + B_x \cdot bias_t + \epsilon) \quad (15)$$

where y refers to confidence ratings, bias can be either 0, $\Sigma Q$, Qc, or $V$ in different models, and $\epsilon$ is the error term (sampled from a Gaussian distribution with zero mean). $\varphi(x)$ is the logistic link function $\varphi(x) = 1/(1+e^{-x})$.

**Confidence models – model optimization and comparison.** Confidence model parameters were estimated by fitting robust linear regression, via the procedure of maximizing log-likelihood (LL), as implemented in MATLAB robustfit functions. Considering that no principled priors for the confidence models are available, we used LL to approximate model evidence for each subject and each model as the BIC (Bayesian information criterion), defined as

$$BIC = nlog(m) - 2LL \quad (16)$$

where $n$ is the number of parameters and $m$ is the number of data points (trials). Similarly to the learning models, we fed the BIC (from each subject in each model) to the random-effect BMS routine implemented in the mbb-vb-toolbox (http://mbb-team.github.io/VBA-toolbox[95]).

Note that parameter recovery and model recovery for the confidence models are also detailed in ref. 32.

**fMRI**
**fMRI acquisition.** The fMRI data were acquired using a 3.0-Tesla Philip Achieva scanner with 32 channels head array coil. We recorded both structural images and functional brain images. T1 weighted structural scans were recorded with the following parameters: FOV (Field of View): $240 \times 180 \times 220\,mm^3$, Voxel size $= 1 \times 1 \times 1\,mm^3$, TR = 8.2 ms and TE = 3.7 ms. Each T2*-weighted functional scan consisted of 36 axial echo-planar images (EPI) acquired in ascending sequence with voxel size of $3 \times 3 \times 3\,mm^3$, slice gap = 0.3 mm, TR= 2000 ms, TE = 28 ms and the flip angle of 76°. Each subject completed three runs in a scanning session. Given the task was self-paced and the fMRI scanner was manually terminated (i.e., ~10 s after the last feedback phase), the total numbers of functional scans for each subject in each run were not the same. Most participants completed the task in around 15 min. The field maps (i.e., magnetic field's inhomogeneity) were collected as well between the second and the third run.

**fMRI preprocessing.** The functional images were preprocessed using SPM12 (Wellcome Department of Imaging Neuroscience, London) with the following steps: realignment and unwarp, co-registration, segmenting anatomical images, normalization, and smoothing. To correct for potential head movement during functional images collection, all functional volumes (from three runs) were realigned to the first volume in the first run and were un-warped with collected field maps. To improve the quality of the following normalization, the mean functional (the output from realignment) and anatomical images were co-registered. The anatomical image from each subject was segmented into six images (i.e., gray matter, white matter, cerebrospinal fluid, fat tissue and air) using nonlinear deformation fields and SPM12's Tissue Probability Maps (TPMs). All segmented images were then normalized to the Montreal Neurological Institute T1 template (i.e., MNI152) using forward deformation fields from the segmentation output. Finally, the EPI images were normalized and smoothed with a full-width half

maximum Gaussian kernel of 6-mm (two times of voxel size of functional images) full-width at half maximum isotropic Gaussian kernel.

**fMRI analysis: GLMs.** Our fMRI analyses leveraged a total of five different GLMs (whose specificities are briefly described below, and summarized in Table 1). All GLMs modeled separately the four main events composing our prototypical trial: symbol presentation, choice, confidence rating, outcome. These event-related regressors were modeled using boxcar functions with corresponding durations. Across all models, the choice and confidence onsets were respectively modulated with parametric modulators accounting for (1) choice (right or left), (2) the distance between initial and final rating point for rating onset. Across all models, to minimize regressor collinearity and to ensure that regression parameters from different conditions and variables were comparable, all parametric modulators were ultimately z-scored (i.e., mean-centered, then standardized to have a standard deviation of 1) at the level of each session of each individual participant[38]. To allow different regressors to fairly compete in explaining the same share of data variance, SPM serial orthogonalization was turned off, and we verified the absence of serious collinearity issues by checking that Variance Inflation Factors remained below conventional, stringent threshold (<5). To remove motion artifact and to improve the quality of fMRI results, all the GLMs also contained six realignment parameters, which were created during preprocessing. Linear contrasts of regression coefficients were designed at the individual level (first-level), and, unless otherwise specified, taken to the group-level random-effect analysis (second-level). For whole brain analyses, second-level analyses consisted of one-sample t-test, whose statistical significance was defined with whole-brain cluster-defining height threshold at uncorrected $p < 0.001$ and family-wise error (FWE)-corrected threshold of $p < 0.05$. Whole-brain statistical tests correspond to one-sided tests of hypotheses. For ROI analyses, the individual-level averaged contrast values were extracted from the ROI using spm built-in function (i.e., spm_get_data.m). These values were then taken to second-level analyses, consisting of one-sample or paired t-tests, as well as two-way repeated-measures ANOVAs. ROI statistical tests correspond to one-sided tests of hypotheses.

GLM1 divided symbol onset and outcome onset into four conditions each (i.e., GP, LP, GC, LC). These eight events of interest were enriched with parametric modulators accounting for 1) confidence ratings for each condition-specific symbol onset, 2) received outcome (coded as 1/ 0 for a relatively good /bad outcome) for each condition-specific outcome phase.

GLM2$_{WID}$ and GLM2$_{SPE}$ featured a single regressor for the symbol and for the outcome events, effectively concatenating all conditions. GLM2$_{WID}$ and GLM2$_{SPE}$ only differed from each other regarding the variable used as the confidence parametric modulator. In GLM2$_{WID}$, confidence consisted in the native ratings. In GLM2$_{SPE}$, confidence ratings were first z-scored *per condition* and before being re-concatenated as a single variable.

GLM3-5 implemented model-based fMRI, and leveraged the latent variables obtained from our winning computational model see "Methods" and Fig. 4a (see also Supplementary Fig. S1). Because the computational variables are meant to capture the difference between conditions, these GLMs also featured a single regressor for the symbol and for the outcome events. As is customary in functional neuroimaging studies, and although beyond the scope of this manuscript, all those GLMs featured the modeled prediction-error (PE) as a parametric modulator of the outcome event.

In GLM3, the symbol presentation onset was modulated by Qc (chosen option value), Qu (unchosen option value), and $V$ (context value).

In GLM4, the symbol presentation onset was modulated by Qc (chosen option value), |Qc-Qu| (absolute value differences), and previous confidence (conf$_{t-1}$).

In GLM5, the symbol presentation onset was modulated by confidence and Qc (chosen option value).

**ROI analyses.** ROIs were created using the marsbar toolbox[96]. A first family of ROIs was built from the significant clusters from the GLM1 confidence activations (VMPFC, dmPFC, Inferior Frontal Gyrus, and Insula).

Alternative VMPFC ROIs were also built from independent meta-analyses[48] and from significant clusters from other analyses of the recent study (e.g., voxels significantly correlated to Qc in GLM3).

**Bayesian model selection (fMRI).** BMS was effectuated using SPM's toolbox: MACS[41]. In the first step (i.e., model assessment), the first-level GLMs of interest from each subject were used to estimate voxel-wise cross-validated log model evidence (cvLME) maps. The maps were generated for each GLM and each subject within the model space. In the second step (i.e., model comparison and selection), the cvLME maps served as inputs for the cross-validated Bayesian Model Selection to compare GLMs within the model space. Only voxels available in all participants were included in those analyses.

### Reporting summary
Further information on research design is available in the Nature Portfolio Reporting Summary linked to this article.

## Data availability
The behavioral data generated in this study have been deposited in OSF [https://osf.io/s92tj/]. The raw MRI data have been deposited in the Donders Repository [https://neurovault.org/collections/MOTXHGZV/]. Source data are provided with this paper.

## Code availability
All Matlab code necessary to reproduce our analyses is available, without restriction at https://osf.io/s92tj/.

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

## Acknowledgements

The authors thank Tiffany M. Hrkalović for assistance with the fMRI data collection. This study was funded by startup funds from the Amsterdam School of Economics awarded to J.B.E. C.C.T. is supported by GSSA, MOE Taiwan Scholarship (1081007012). M.L. is supported by an ERC Starting Grant (INFORL-948671). S.P. is supported by an ERC Consolidator Grant (RaReMem-101043804) and by the Agence National de la Recherche (CogFinAgent: ANR-21-CE23-0002-02; RELATIVE: ANR-21-CE37-0008-01; RANGE: ANR-21-CE28-0024-01).

## Author contributions

C.C.T., S.P., J.B.E. and M.L. designed the study. JBE acquired funding. C.C.T. ran the experiment under the guidance of J.B.E. C.C.T. conducted the analysis, and drafted the manuscript under the supervision of M.L. N.S.G. performed the model-validation analyses. All authors (C.C.T., N.S.G., S.P., J.B.E. and M.L.) critically assessed and discussed the results, and revised and approved the manuscript.

## Competing interests

The authors declare no competing interests.
