## [Peer Review File · Nature Communications]

Neural and computational underpinnings of biased confidence in human reinforcement learningREVIEWER COMMENTS

Reviewer #1 (Remarks to the Author):

In this paper, Ting et al. seek to test whether two prefrontal networks associated with confidence (a positive and a negative network given the sign of the correlation) are also functionally dissociated, i.e. whether they differentially contribute to the computations of confidence. To this end, the authors leverage an established reinforcement learning task that produces confidence biases while matching overall learning accuracy. The authors separate confidence in multiple signals: objective uncertainty, task-wide confidence, and condition-specific confidence. This work first replicates previous behavioral findings on valence-induced confidence bias. Through standard GLM as well as model-based analyses of fMRI data, the authors report that DLPFC/DMPFC (negative network) to correlate with condition-specific confidence, and the VMPFC (positive network) to correlate with absolute confidence. Finally, and most strikingly, the author report that the VMPFC activity is, in fact, better explained by confidence, and not value as generally assumed in the field.

Overall this is a well written manuscript that I enjoyed reading. It is timely, the question of broad interest, and touches on a topic that deserves more attention.

While I am positive on the potential conceptual advance and the strength of using reinforcement learning to better understand the neural computations of confidence, I have serious concerns about the validity of the claims in light of the analyses, results and possible experimental confounds, which I am not sure can be easily addressed.

That is, some experimental variables used for the fMRI analyses appear confounded, such as the condition-specific confidence, which should inevitably correlate with accuracy. Any neural finding may simply/partly reflect changes in accuracy, rather than purely confidence.

Other results are inconclusive: in figure 3 the key result should be the interaction between the contexts and ROIs, and between confidence measures (GLM2_wid and GLM2_spe) and ROIs. However, these interactions are not reported.

Again in figure 3, especially panel C, I would not consider the results as converging: if anything, I find it odd that one set of analysis gives strong evidence for one result, which then becomes null in the second analysis. This suggests potential issue(s) in the analysis (and/or experimental design).

Absolute confidence (as used in GLM2_wid) should also be z-scored / mean-centered across the whole experiment (not done it seems), while in GLM2_spe z-scoring should be done per condition (done). Parametric modulators should always be mean-centered, as collinear regressors are highly problematic. This comment applies to all parametric modulations in the various GLMs of this work, as I am not sure when/where mean-centering/z-scoring was done.

Figure 6B: all ROIs, not only VMPFC, seem to be involved in value and confidence processing? The DMPFC and IFG+INS show the same results (just different signs). I am not sure I understand what the

authors meant in p21, “BOLD signal in the VMPFC correlates with confidence-building variables”, because all selected ROIs do.

Finally, for the last key analysis and result (figure 7), I imagine confidence and value are correlated (I would like to see the correlation between confidence and value). The modelling itself suggests that the best model has bias term corresponding to chosen value Q_c , and the parameter value is large (fig 5B) and this provides a great fit to participants’ behavioral data (p27 discussion). The GLM analysis therefore could well be confounded: although SPM usually deals gracefully with moderately correlated regressors / modulators, high correlations prevent proper inference. This could be the cause for the inconclusive results of BMC for different models (only value, only confidence, both) as reported by the authors (p22). Thus, I am not convinced by this analysis and the conclusions that can be drawn.

In sum, I feel several key neuroimaging analyses and ensuing results are confounded, which means at present it is not possible to draw clear (and strong) conclusions.

Minor points:

confidence incentive method. Although this might be a well-acceptable method, it seems asymmetric: favoring high confidence ratings in correct trials but less so for low confidence in error trials. It would be good to see the accuracy in the task broken down by confidence levels (e.g. quartiles).

Figure 3B, it was a bit confusing that the sign of the betas cue-evoked intercept are negative for positive network and positive for the negative network.

Abstract (but also intro): “humans and animals constantly produce subjective confidence judgments”, that is arguably not true. People (and other animals) keep track of their performance, thoughts, etc, but seldom through explicit confidence judgements.

personal opinion, stat results can be reported at the 2nd decimal (or 2nd value after 0 in p-values e.g. $p=0.0351$ as $p=0.035$ or $p=0.04$)

Reviewer #2 (Remarks to the Author):

In the present study, the authors leverage a well-established confidence bias, higher confidence for gains compared to losses, to functionally dissociate two networks that typically track confidence. Confidence typically correlates positively with a ventromedial prefrontal network and negatively with a dorsolateral and dorsomedial prefrontal network. They use a reinforcement learning paradigm that manipulates the context (gain vs loss) and the informativeness of feedback (partial vs full feedback) and fMRI to test the hypothesis that VMPFC integrates biases (e.g. gain vs loss) whereas DMPFC is more sensitive to uncertainty (e.g. for partial vs full feedback). Confirming the first part of their hypothesis, the authors find that the former network integrates confidence biases, whereas the latter does not. At odds with the second half of their hypothesis, they do not find that the latter network is sensitive to partial vs full information.

As a control, the authors conduct model-based analyses to rule out that the VMPFC merely codes for option values instead of confidence.

This work is interesting, and thoroughly performed. I don't have many comments, however, I would like the authors to include a brief discussion of a recent paper by Trudel et al. showing both positive and negative effects of uncertainty in VMPFC. How does that square with the authors' findings and interpretation?

I also think it would help the readability of the paper if the authors reduced the number of acronyms they are using.

Trudel, N., Scholl, J., Klein-Flugge, M. C., Fouragnan, E., Tankelevitch, L., Wittmann, M. K., & Rushworth, M. F. S. (2020). Polarity of uncertainty representation during exploration and exploitation in ventromedial prefrontal cortex. *Nat Hum Behav*. [<https://doi.org/10.1038/s41562-020-0929-3>]

REVIEWER COMMENTS

Reviewer #1 (Remarks to the Author):

In this paper, Ting et al. seek to test whether two prefrontal networks associated with confidence (a positive and a negative network given the sign of the correlation) are also functionally dissociated, i.e. whether they differentially contribute to the computations of confidence. To this end, the authors leverage an established reinforcement learning task that produces confidence biases while matching overall learning accuracy. The authors separate confidence in multiple signals: objective uncertainty, task-wide confidence, and condition-specific confidence. This work first replicates previous behavioral findings on valence-induced confidence bias. Through standard GLM as well as model-based analyses of fMRI data, the authors report that DLPFC/DMPFC (negative network) to correlate with condition-specific confidence, and the VMPFC (positive network) to correlate with absolute confidence. Finally, and most strikingly, the author report that the VMPFC activity is, in fact, better explained by confidence, and not value as generally assumed in the field.

Overall this is a well written manuscript that I enjoyed reading. It is timely, the question of broad interest, and touches on a topic that deserves more attention.

While I am positive on the potential conceptual advance and the strength of using reinforcement learning to better under the neural computations of confidence, I have serious concerns about the validity of the claims in light of the analyses, results and possible experimental confounds, which I am not sure can be easily addressed.

We thank R1 for acknowledging that our study is timely and of broad interest, and for sharing their concerns about our analyses, results and some possible experimental limitations. As we show below in our detailed response, the validity of our claims is robust to each of those concerns. We believe that the additional analyses provided in this rebuttal have significantly improved our manuscript and raised confidence in our findings.

That is, some experimental variables used for the fMRI analyses appear confounded, such as the condition-specific confidence, which should inevitably correlate with accuracy. Any neural finding may simply/partly reflect changes in accuracy, rather than purely confidence.

R1 is concerned that fMRI activities that we relate to confidence are confounded by accuracy.

- Before we report additional analyses inspired by the reviewer's comment, we would like to consider the conceptual level. We note that the idea that a neural (internal) variable tracks the objective (external) accuracy (i.e. whether a response is correct or not) *before the choice is implemented and feedback received* (which is when all our fMRI analyses of interest are performed) is challengeable: how would these external contingencies be known and integrated in the brain before the choice is made? If this variable does exist in the brain, why is it not used to perform the task at perfect accuracy? If we relax the definition, and consider that accuracy could be represented as a subjective approximation of the probability of being correct conditioned by the available evidence, then this new definition literally corresponds to our (and others') definition (and operationalization through our incentivized probabilistic rating) of confidence (Pouget et al., 2016).

To make these features clearer; we adjusted **the Figure 1** (see reproduction below) to highlight that fMRI analyses of confidence largely took place during the symbol presentation by highlighting this period with a yellow, transparent box. We believe that this better convey that our task design is ideal for our purpose, not only because participants do not yet have information about which option is the correct one at the time-point when we analyze confidence signals, but also because they do not even know how they are going to implement their choice.

Updated Figure 1. This Figure now features a yellow box to highlight when our analysis of confidence signals takes place.

- Second, we note that the presence of a main effect of valence (gain vs loss) demonstrated in the VMPFC goes against the idea that the confidence-related activations are confounded by accuracy, given that confidence is affected by valence, whereas accuracy is identical in gain and loss contexts (the so-called valence-induced bias).

Despite those conceptual objections, we found R1's suggestion stimulating, and engaged in a set of control analyses.

We designed a new $GLM2_{ACC}$, similar to $GLM2_{WID}$, where we replaced the parametric modulator of confidence with trial-by-trial accuracy (also Z-scored at the individual level)¹. Interestingly, at the whole brain level and conventional statistical thresholds ($P < 0.001$ uncorrected voxel-wise, $P < 0.05$ FWE-corrected at the cluster level; $k = 47$), significant negative effects of accuracy were found in the DMPFC and bilateral IFG, but no significant activations were found in the VMPFC (**Figure R1A**).

Figure R1 | Positive and negative networks better encode confidence than accuracy. (A) Whole brain activations with positive (left) and negative accuracy (right) estimated from $GLM2_{ACC}$. Activations are corrected for multiple comparisons at the cluster level ($P < 0.001$ uncorrected voxel-wise, $P < 0.05$ FWE-corrected at the cluster level). (B) ROI analyses, comparing regression coefficients estimated independently for accuracy ($GLM2_{ACC}$) and confidence ($GLM2_{WID}$ and $GLM2_{SPE}$). (C) ROI analyses, comparing regression coefficients estimated for accuracy and confidence in the same GLM ($GLM6_{WID}$ and $GLM6_{SPE}$).

Histograms and error bars represent means \pm SEM of activations. Dots represent individual estimates. ***: $P < 0.001$; **: $P < 0.01$; *: $P < 0.05$; ~: $0.05 < P < 0.10$.

Using the anatomical VMPFC ROI from Bartra, et al 2013, we extracted the regression coefficients for confidence in $GLM2_{WID}$ and for accuracy in $GLM2_{ACC}$ (regressors were both Z-scored in the design matrix, for commensurability). A direct comparison confirmed that BOLD signal in the VMPFC barely encodes accuracy, with an association strength

¹ Note that an equivalent of GLM1 where we replace confidence with accuracy failed to converge, as several sessions featured 100% correct response in some contexts, rendering the parametric modulator non-estimable.

significantly below the one observed with confidence ($t_{37} = -5.16$; $P = 8.65 \times 10^{-6}$; **Figure R1B, left**)

For the negative network, we defined our ROI as the conjunction of the clusters correlating negatively with confidence in GLM2_{SPE} and negatively with accuracy in GLM2_{ACC}. We then extracted the regression coefficients for confidence in GLM2_{SPE} and for accuracy in GLM2_{ACC} in these *unbiased* ROIs (again, both regressors being Z-scored in the design matrix, for commensurability). A direct comparison confirmed that BOLD signal in both regions of the negative network encodes accuracy, but significantly less than confidence (DMPFC: $t_{37} = -2.95$; $P = 0.006$; IFG+INS: $t_{37} = 2.55$; $P = 0.015$; **Figure R1B, middle & right**),

To complete those analyses, we decided to design GLMs where confidence and accuracy regressors could directly compete to explain variance (all regressors Z-scored, and SPM orthogonalization turned-off). We therefore designed 2 new GLMs, GLM6_{WID} and GLM6_{SPE}, which respectively featured, as parametric regressors, accuracy and task-wide confidence (GLM6_{WID}) and accuracy and context-specific confidence (GLM6_{SPE}). These GLMs allowed us to test if our attribution of task-wide confidence to the positive network and of context-specific confidence to the negative network holds in the presence of a potential accuracy confound. In both cases, confidence regression coefficients significantly outmatched accuracy regression coefficients (VMPFC: $t_{37} = -5.22$; $P = 1.14 \times 10^{-5}$; DMPFC: $t_{37} = 1.96$; $P = 0.058$; IFG+INS: $t_{37} = 1.94$; $P = 0.060$; **Figure R1C**)

Altogether these analyses unambiguously confirm our claims. Not only did our results replicate while controlling for the presence of the accuracy confound, but the dissociation between the positive and negative network is consolidated by these additional analyses. Indeed, the analyses suggest that VMPFC is less correlated with accuracy than the negative network, consistent with the idea that it incorporates additional contextual information, such as the valence-induced bias that is the focus of our current experiment.

Other results are inconclusive: in figure 3 the key result should be the interaction between the contexts and ROIs, and between confidence measures (GLM2_wid and GLM2_spe) and ROIs. However, these interactions are not reported.

Actually, it is not standard, in fMRI analysis, to compare activation measures (betas) between regions, notably because the hemodynamic response function and its sensitivity to different stimuli and processes can vary across regions, potentially creating apparent but misleading differences. Also, brain regions or voxels are generally not conceptualized as a control of one another: for instance, when one reports whole brain activations, those activations are tested in each voxel independently, not in relation to neighboring voxels. The erroneous analyses of interactions in neuroscience (Nieuwenhuis et al., 2011) applies to situations where experimental designs feature control and test conditions, and one fails to analyze the test conditions with respect to the control condition (which is not the case here).

Despite these principled objections, we nonetheless considered R1's suggestion.

First, we tested the effect of ROI on the main effect of our experimental factors depicted in **Figure 3B** (valence, information, interaction). To do so, we used the properties of our linear design, and combined our GLM1 activations across conditions (as a reminder, our conditions were valence: (G)ain/(L)oss and information: (P)artial/(C)omplete; valence: [GP + GC] – [LP + LC]; information: [GC + LC] – [GP + LP]; interaction: [GP + LC] – [GC + LP]), to obtain one activation per effect, participant and ROI. We then performed one-way repeated-measure

ANOVAs, to test the effect of ROI independently for each factor. We got a significant effect of ROI for valence ($F_{2,35} = 10.83$; $P = 5.49 \times 10^{-5}$) and for the interaction ($F_{2,35} = 4.41$; $P = 0.0156$), but not for information ($F_{2,35} = 0.93$; $P = 0.40$). For both significant results, the ROI effect is driven by the VMPFC effects being significant and different from the other regions (see **Figure R2A**).

Second, we tested the effect of ROI on the differential activation to task-wide versus confidence-specific variables, depicted in **Figure 3C**. We simply computed an index of the difference between these two regressors ($[GLM2_{WIDConf}] - [GLM2_{SPEConf}]$) per participant and ROI, and then performed a one-way repeated-measures ANOVA to test the effect of ROI on this differential activation. Again, we got a significant main effect of ROI ($F_{2,35} = 22.64$; $P = 2.12 \times 10^{-8}$), driven by the VMPFC effects being significant and different from the other regions (see **Figure R2B**).

Figure R2 | Testing the main effect of ROI on experimental factors (A) and confidence encoding (B). (A) We combined our GLM1 activations across conditions (valence (left): [GP + GC] – [LP + LC]; information (middle): [GC + LC] – [GP + LP]; interaction (right): [GP + LC] – [GC + LP]), to obtain one activation per effect, participant and ROI. We then performed one-way repeated-measures ANOVAs, to test the effect of ROI independently for each factor. (B) We computed an index of the difference between these two regressors ($[GLM2_{WIDConf}] - [GLM2_{SPEConf}]$) per participant and ROI, and then performed a one-way repeated-measure ANOVAs, to test the effect of ROI on this differential activation. Histograms and error bars represent means \pm SEM of activations. The figure depicts the result of ANOVAs (red) and post-hoc tests estimated from Matlab's multcompare function (black).
 ***: $P < 0.001$; **: $P < 0.01$; *: $P < 0.05$; ~: $0.05 < P < 0.10$.

Overall, these results unambiguously confirm our claim of dissociation between the positive network (VMPFC) and the negative network. Yet, because of the conceptual objection outlined above, we chose not to include these new set of analyses in the Main Text. If R1 and/or the editor think otherwise and deem it necessary, we will be happy to integrate them (e.g. in the supplementary material).

Again, in Figure 3, especially panel C, I would not consider the results as converging: if anything, I find it odd that one set of analysis gives strong evidence for one result, which then becomes null in the second analysis. This suggests potential issue(s) in the analysis (and/or experimental design).

While we understand R1's concern that the comparison of regression coefficients and the Bayesian Model Comparison (BMC) do not provide the exact same picture of the data in

Figure 3, we note that this is actually not uncommon and definitely not an indication of an issue in the analysis. For instance, take the example of a variable which would be perfectly encoded in a ROI, but positively in half of the subjects, and negatively in the other half (with the same absolute effect size): a random-effect analysis (t-test) on the regression coefficient would be non-significant, with an estimated effect-size of 0 (because activations cancelled each other in the 2 sub-populations of subjects); yet, because the variable explains 100% of the ROI variance, a BMC analysis would highly favor any model including the said variable.

These two types of analyses should therefore be seen as complementary, and we believe it is a good practice to report on both, even when they converge to slightly different conclusions. Providing the information allows the reader to draw their own informed inferences about our results based on their preferred analysis approach. Besides, note that in our case, both conclusions, though arguably different, are actually in line with our main claim, and all our other analyses: VMPFC seems to preferentially encode a variable akin to task-wide confidence, while the negative network seems to preferentially encode a variable akin to context-specific confidence.

Absolute confidence (as used in GLM2_wid) should also be z-scored / mean-centered across the whole experiment (not done it seems), while in GLM2_spe z-scoring should be done per condition (done). Parametric modulators should always be mean-centered, as collinear regressors are highly problematic. This comment applies to all parametric modulations in the various GLMs of this work, as I am not sure when/where mean-centering/z-scoring was done.

We definitely agree with R1 on this. Actually, all our parametric regressors are z-scored across the whole experiment (i.e. mean-centered then standardized to have a standard deviation of 1), as originally stated in the methods. To prevent any misunderstanding, we improved this methods section as follows (on page 37):

Across all models, to minimize regressor collinearity and to ensure that regression parameters from different conditions and variables were comparable, all parametric modulators were ultimately z-scored (i.e. mean-centered, then standardized to have a standard deviation of 1) at the level of each session of each individual participant, (Lebreton, et al., 2019).

Note that in the case of GLM2_spe, this whole-experiment Z-scoring was implemented after –and in addition to– the session-specific Z-scoring.

To improve readability and alleviate readers potential concerns early in our manuscript, we also added this information in the Results section, when presenting the first fMRI results (GLM1; on page 11):

note that, to ensure between-subject and between-regressor commensurability, all parametric modulators of all fMRI GLMs were z-scored at the session and individual level (Lebreton, et al., 2019)

Figure 6B: all ROIs, not only VMPFC, seem to be involved in value and confidence processing? The DMPFC and IFG+INS show the same results (just different signs). I am not sure I understand what the authors meant in p21, “BOLD signal in the VMPFC correlates with confidence-building variables”, because all selected ROIs do.

R1 is correct in highlighting that, in **Figure 6B** the DMPFC and IFG+INS also seem to correlate with additional confidence variables, above and beyond Qc (notably, most robustly $conf_{t-1}$). Yet, for those analyses (and the following ones) we decided to mostly focus on VMPFC in the Main Text (while still reporting results for the DMPFC and IFG+INS in **Figure 6B** for

completeness and transparency) for two main reasons. First, in the past literature, mostly the VMPFC has been quasi-systematically associated with the encoding of Q_c during reinforcement-learning. Hence, the set of analyses attempting to dissociate Q_c and confidence are mostly relevant for this region. Second, from **Figure 6A**, it was already evident that DMPFC and IFG+INS do not exclusively encode Q_c , but also Q_u (contrary to VMPFC). Hence, as we discuss in the manuscript, these regions are more probably involved in the comparison of available options (including the confidence associated with this comparison), and the set of analyses attempting to dissociate Q_c and confidence naturally become less relevant for these regions.

We now allude to this reasoning in the revised version of the Main Text (on page 20):

*Though regions of the negative network (DMPFC and IFG+INS) also seem to correlate with additional confidence variables, above and beyond Q_c (notably and most robustly $conf_{t-1}$; DMPFC: $t_{37} = 3.46$, $P = .0014$ and IFG+INS: $t_{37} = -4.32$, $P = .0001$; **Figure 6B**), the set of analyses dissociating Q_c from confidence encoding seems less relevant there. Indeed, the negative network has been less systematically associated with value encoding in previous studies. In addition, analyses reported in **Figure 6A** already show that signal in these regions not only correlates with the value of the chosen option (Q_c), but also robustly (with an opposite sign) with the value of the unchosen option (Q_u). This pattern is tentatively consistent with a role in the comparison of available options (rather than valuation), including potentially the context-specific confidence associated with this comparison.*

Finally, for the last key analysis and result (figure 7), I imagine confidence and value are correlated (I would like to see the correlation between confidence and value). The modelling itself suggests that the best model has bias term corresponding to chosen value Q_c , and the parameter value is large (fig 5B) and this provides a great fit to participants' behavioral data (p27 discussion). The GLM analysis therefore could well be confounded: although SPM usually deals gracefully with moderately correlated regressors / modulators, high correlations prevent proper inference. This could be the cause for the inconclusive results of BMC for different models (only value, only confidence, both) as reported by the authors (p22). Thus, I am not convinced by this analysis and the conclusions that can be drawn.

The reviewer is correct in inferring that confidence and value are correlated. At the session level (which is the level of description relevant for the fMRI analysis), the correlation between confidence and Q_c is 0.49 ± 0.02 . To further evaluate if this correlation creates concerning multicollinearity issues, we computed the Variance Inflation Factor characterizing the impact of this correlation on regression parameter estimates, using both an in-house Matlab code and an external code provided in an fMRI processing toolbox (the CANlab toolbox: <https://github.com/canlab/CanlabCore>). Both methods converged on identical estimates, which are significantly below concerning threshold (average VIF estimates in GLM5: 1.46 ± 0.059 ; Max/min: 2.61/1.01; stringent multicollinearity VIF threshold: 5). We now allude to these verifications (page 37):

and we verified the absence of serious collinearity issues by checking that Variance Inflation Factors remained below conventional, stringent threshold (< 5).

Also, note that the results of the analysis of GLM5 (namely, that activations in VMPFC are better explained by confidence than by Qc) are supported by other analyses devoid of such confidence/Qc correlation “issues”. Indeed, the results provided in **Figure 6B** clearly illustrate that, in addition to Qc, VMPFC activity correlates with other variables that form confidence, such as $|Qc - Qu|$ and $conf_{t-1}$.

In summary, we believe that our analyses robustly support our claims, despite the existence of correlations between Qc and confidence.

In sum, I feel several key neuroimaging analyses and ensuing results are confounded, which means at present it is not possible to draw clear (and strong) conclusions.

We thank R1 for giving us the possibility to clarify and expand on our findings. We believe that the set of additional analyses provided in this revision in response to R1’s concerns show that our analyses are not critically confounded and that, as a consequence, our conclusions remain valid and robust.

Minor points:

confidence incentive method. Although this might be a well-acceptable method, it seems asymmetric: favoring high confidence ratings in correct trials but less so for low confidence in error trials. It would be good to see the accuracy in the task broken down by confidence levels (e.g. quartiles).

This method (matching probability: MP) is a well validated method originally developed by psychologists and further refined by behavioral economists who demonstrated that it is fully incentive compatible, over a wide range of assumptions and utility functions (Ducharme & Donnell, 1973; Hollard et al., 2016; Karni, 2009; Schlag et al., 2015; Schotter & Trevino, 2014). This means that the optimal strategy of a participant is to report their confidence, as a subjective estimate of the probability of being correct, as accurately and truthfully as possible. Thus, if participants are metacognitively sophisticated and perform the task correctly, it is a desirable property that the MP mechanism incentivizes them to report high confidence when they believe they have a high probability of being correct, and low confidence when they believe they have a low probability of being correct.

More generally, the MP mechanism is not asymmetric, and ensure that the loss function around the ideal confidence rating is similar across the whole response scale. We actually illustrated this point in a previous article (Lebreton et al., 2018), which we reproduce here as **Figure R3**. The same paper contains the mathematical demonstration that loss functions are quadratic (hence are symmetric).

Figure R3 | Incentive-compatibility of the Matching probability mechanism, from Lebreton, et al. (2018). We simply computed an index of the Expected probability of winning $E(x)$ induced by the MP mechanism, as a function of the chosen rating x for several levels of underlying “true” confidence c (colour scale). The dots indicate the highest point of each curve, which correspond to $x = c$. This illustrates that in all cases, the optimal rating (x) corresponds to the true confidence (c), and that the loss function (i.e. decrease in $E(x)$ around the optimal rating x) is quadratic and symmetrical toward over- and under-confidence.

Figure 3B, it was a bit confusing that the sign of the betas cue-evoked intercept are negative for positive network and positive for the negative network.

We thank R1 for noting this potentially confusing pattern. We added a clarification sentence in the caption of **Figure 3B** (page 12).

Note that the notion of positive versus negative network characterizes the sign of the correlation of activations with confidence. In the present panel, cue-evoked activity exhibits an opposite pattern, with negative baseline activations in the positive network, and positive baseline activations in the negative network.

Abstract (but also intro): “humans and animals constantly produce subjective confidence judgments”, that is arguably not true. People (and other animals) keep track of their performance, thoughts, etc, but seldom through explicit confidence judgements.

Although we understand the reason behind R1’s comment, we note that the notion of confidence judgments is routinely used for the *implicit* tracking of an agent’s own performance, notably also when this agent is a rodent – see e.g. (Kepecs & Mainen, 2012). We nonetheless amended our abstract and introduction as suggested by R1, to avoid any misconception.

personal opinion, stat results can be reported at the 2nd decimal (or 2nd value after 0 in p-values e.g. $p=0.0351$ as $p=0.035$ or $p=0.04$)

Thank you for sharing this. Conventions for reporting p values diverge across journals and we decided to continue with the convention of reporting 4 decimals for all p-values as this provides higher accuracy. We are happy to change this if the editor and reviewer insist.

Reviewer #2 (Remarks to the Author):

In the present study, the authors leverage a well-established confidence bias, higher confidence for gains compared to losses, to functionally dissociate two networks that typically track confidence. Confidence typically correlates positively with a ventromedial prefrontal network and negatively with a dorsolateral and dorsomedial prefrontal network. They use a reinforcement learning paradigm that manipulates the context (gain vs loss) and the informativeness of feedback (partial vs full feedback) and fMRI to test the hypothesis that VMPFC integrates biases (e.g. gain vs loss) whereas DMPFC is more sensitive to uncertainty (e.g. for partial vs full feedback). Confirming the first part of their hypothesis, the authors find that the former network integrates confidence biases, whereas the latter does not. At odds with the second half of their hypothesis, they do not find that the latter network is sensitive to partial vs full information.

As a control, the authors conduct model-based analyses to rule out that the VMPFC merely codes for option values instead of confidence.

This work is interesting, and thoroughly performed. I don't have many comments, however, I would like the authors to include a brief discussion of a recent paper by Trudel et al. showing both positive and negative effects of uncertainty in VMPFC. How does that square with the authors' findings and interpretation?

We thank R2 for evaluating our study as interesting and thoroughly performed.

The Trudel et al. study indeed suggests that the coding of uncertainty (i.e. inverse confidence) in the VMPFC can reverse polarity – switching from positive during initial exploration, to negative during exploitation. In our taxonomy, that would mean that VMPFC is first part of the negative network (because our reference is confidence rather than uncertainty) and then gradually switches to being part of the positive network.

Several features, however, significantly differ between their setups and ours, which make cross-interpretations difficult: information seeking vs instrumental learning tasks, blocked vs interleaved design, only partial vs partial + complete information, variable & explicit vs ambiguous & fixed horizon, etc.

We speculate that our simple instrumental-learning task –which notably features fixed outcome contingencies, interleaved contexts and 50% of complete information trials–, primes participants to behave myopically, i.e. to maximize their earnings at each trial, without strategizing about information seeking and exploration. This seems confirmed by the very strong and robust *confirmatory updating* aspect of our winning RL model, which shows and explains how participants tend to quickly build –and stick to– their current strategies. These elements seem to indicate that there is little exploration going on in our task, and that participants tend to mostly behave in an exploitation mode, where the results of Trudel and colleagues are consistent with ours.

Without mentioning these speculations, we now allude to this debate in the discussion on page 27:

A recent study even suggests that, in a task where participants must form beliefs about the accuracy of reward information cues by trial-and-error, the polarity of uncertainty (i.e. inverse confidence) encoding in the VMPFC could reverse, from positive during exploration to negative during exploitation (Trudel et al., 2021). In our taxonomy, that would mean that VMPFC can be first part of the negative network (because our reference is confidence rather than uncertainty) and can then gradually switch to being part of the positive network. Further research should identify under which conditions the polarity of confidence signals in the VMPFC could possibly change.

I also think it would help the readability of the paper if the authors reduced the number of acronyms they are using.

We thank the reviewer for this remark. We kept the acronym to the minimum, notably in the introduction and the discussion where we restricted ourselves to standard ones (VMPFC, BOLD, fMRI...).

Trudel, N., Scholl, J., Klein-Flugge, M. C., Fouragnan, E., Tankelevitch, L., Wittmann, M. K., & Rushworth, M. F. S. (2020). Polarity of uncertainty representation during exploration and exploitation in ventromedial prefrontal cortex. **Nat Hum Behav**.

[<https://doi.org/10.1038/s41562-020-0929-3>]

REVIEWERS' COMMENTS

Reviewer #1 (Remarks to the Author):

I commend the authors for a strong rebuttal and strengthened paper. They have taken the time to propose well reasoned arguments as well as several new analyses that answer the concerns raised. I am happy with the revised manuscript and can

congratulate the authors for this interesting and thought-provoking work.

Reviewer #2 (Remarks to the Author):

Thank you for unpacking how the studies relate and adding a cue to the debate for the reader. Thank you also for reducing the number of acronyms to make the manuscript easier to read. I believe this manuscript is now ready for publication. Congratulations to the authors on a nice contribution to the literature.